# Integrating Drug Substructures and Longitudinal Electronic Health Records for Personalized Drug Recommendation

**Wenjie Du**[1,2], **Xuqiang Li**[1,2], **Jinke Feng**[1,2], **Shuai Zhang**[1,2,4], **Zhang Wen**[3], **Yang Wang**[1,2,†]

[1]University of Science and Technology of China, China
[2]Suzhou Institute for Advanced Research, USTC, China
[3]Huazhong Agricultural University    [4]Baidu, Inc

{jinkefeng, shuaizhang, xuqiangli}@mail.ustc.edu.cn
zhangwen@mail.hzau.edu.cn   {duwenjie,angyan}@ustc.edu.cn

## Abstract

Drug recommendation systems aim to identify optimal drug combinations for patient care, balancing therapeutic efficacy and safety. Advances in large-scale longitudinal EHRs have enabled learning-based approaches that leverage patient histories such as diagnoses, procedures, and previously prescribed drugs, to model complex patient-drug relationships. Yet, many existing solutions overlook standard clinical practices that favor certain drugs for specific conditions and fail to fully integrate the influence of molecular substructures on drug efficacy and safety. In response, we propose **SubRec**, a unified framework that integrates representation learning across both patient and drug spaces. Specifically, SubRec introduces a conditional information bottleneck to extract core drug substructures most relevant to patient conditions, thereby enhancing interpretability and clinical alignment. Meanwhile, an adaptive vector quantization mechanism is designed to generate patient–drug interaction patterns into a condition-aware codebook which reuses clinically meaningful patterns, reduces training overhead, and provides a controllable latent space for recommendation. Crucially, the synergy between condition-specific substructure learning and discrete patient prototypes allows SubRec to make accurate and personalized drug recommendations. Experimental results on the real-world MIMIC III and IV demonstrate our model's advantages. The source code is available at https://DrugRecommendation/.

## 1 Introduction

Drug recommendation is a pivotal task in healthcare, focused on identifying the optimal combination of drugs to address a patient's diagnosed conditions [26, 40, 8]. This task aligns conceptually with sequential recommendation systems, where decisions are made iteratively across a patient's series of clinical visits. With the increasing availability of individual medical data, such as longitudinal electronic health records (EHRs) [11, 18]—which capture patients' historical visit sequences, including diagnoses, procedures, and prescribed medications—there is a rich foundation for developing learning-based predictive models. In this context, drug recommendation emerges as a critical data mining challenge. It leverages advanced machine learning techniques, particularly deep neural networks, to analyze complex clinical event sequences. By integrating a patient's current clinical events with their historical records, these systems aim to generate personalized drug plans while minimizing drug-drug interactions (DDIs) to satisfy safety principles [28, 7, 19].

---

† : corresponding author

39th Conference on Neural Information Processing Systems (NeurIPS 2025).

Building upon this, EHRs serve as a crucial basis for the recommendations made by the model. Early works typically focused on a patient's individual EHRs, using deep learning models to uncover the intrinsic relationships between a patient's health conditions and prescribed medications. For example, 4SDrug [27] constructed historical visit sets and proposed measuring the similarity between symptom and drug sets for drug recommendation. COGNet [35] developed a novel copy-or-predict mechanism, predicting whether a drug should be copied from historical recommendations or a new drug combination should be suggested. However, from a healthcare provider's perspective, the primary focus is often on the clinical case rather than individual differences [35]. For instance, in the case of alcohol dependence, doctors typically prescribe Naltrexone, whereas for pneumonia, Amoxicillin is commonly prescribed, without the need to consider personal variations, as shown in Figure 1 (a). Therefore, similarity in current patient visits is more common than similarity in the complete medical histories of patients [35, 33]. Representations of patient cases would aid in drug recommendation outcomes, accelerating the process of drug identification [9, 21, 44]. However, directly modeling full patient histories would significantly increase memory and computation costs due to the vast and diverse nature of clinical records. Therefore, adopting a condensed and reusable reference paradigm during the recommendation process is a worthwhile consideration in real applications.

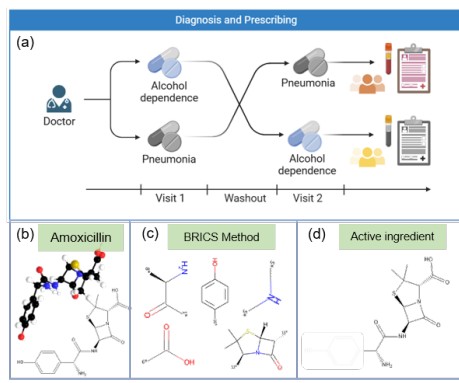

Figure 1: Doctor's Diagnosis and Prescribing Process. (a) Doctors often prescribe Naltrexone for Alcohol Dependence and Amoxicillin for Pneumonia, even for different patients. This is guided by professional medical knowledge. (b) 3D molecular structure of Amoxicillin. (c) The five molecular scaffolds of Amoxicillin, derived using the BRICS method [3]. (d) The $\beta$-lactam ring, the active ingredient of Amoxicillin.

Furthermore, researchers have gradually realized that the biochemical activities of a drug are often linked to a few privileged molecular substructures [14, 17, 45]. Therefore, the active ingredient of a drug can be associated with the function of specific substructures within the drug molecules, and making prescriptions based on molecular substructure awareness may lead to more desirable efficacy and explainability [4, 17, 15]. For example, MoleRec [40] adopts the Break Retrosynthetically Interesting Chemical Substructures (BRICS) method [3] to decompose drug molecules into substructures. SafeDrug also employs the BRICS method to split chemical substructures and is equipped with a global message passing neural network and a local bipartite learning module to fully capture the connectivity and functionality of drug molecules. These methods decompose drugs into several substructures to learn the relationships between case representations. However, directly using rule-based decomposition may disrupt the effective substructure connectivity in drugs, as shown in Figure 1 (b)-(d). Drug activity is typically achieved through the combination of multiple substructures rather than a single one. Moreover, the extraction of substructures should consider the patient's health condition, as other comorbidities may also influence prescription decisions, which is a crucial prerequisite.

In light of this, we propose **SubRec**, a deep learning framework that integrates drug **sub**structures with longitudinal electronic health records (EHRs) for precise and interpretable drug **rec**ommendation. SubRec is designed to address two key challenges in personalized medicine: (1) the complexity and sparsity of patient historical records, and (2) extracting patient-specific core substructures. To handle the vast and diverse nature of patient cases, we extend the idea of vector quantization (VQ) [31, 23] by constructing an adaptive codebook that clusters heterogeneous patient–drug interactions into a compact set of prototypes. This quantized representation avoids the instability of variational modeling in high-dimensional latent spaces, and yields a lightweight, discrete structure that supports efficient similarity matching across patient representations. On the drug side, SubRec improves conditional information bottleneck (CIB) to extract condition-specific core substructures by treating the patient health context as a conditional variable and the molecular graph as the prediction target. This results in concise, interpretable drug embeddings that capture substructures most relevant to a given clinical state. These representations are then aligned with the patient's prototype for personalized drug recommendation. The quantization process avoids the complexity and instability of modeling variational distributions in high-dimensional latent spaces, making the model lighter and more stable during training [37, 46]. The discrete structure of the latent space naturally supports the CIB principle

by enforcing a compact representation [34]. Through the joint modeling of condition-aware drug substructures and quantized patient states, SubRec effectively balances predictive accuracy and interpretability. This synergy represents a novel mechanism for learning discrete, clinically grounded representations, enabling reliable and safe decision-making in drug recommendation tasks.

# 2 Preliminaries

## 2.1 Problem Formulation

Based on the patient's visit history, the model aims to predict the optimal drug combination for the current visit, balancing both accuracy and safety. Accuracy is defined as the alignment between the predicted medications and those prescribed by the doctor. Safety is assessed by the DDI rate.

**Electronic Health Records (EHRs).** EHRs of a patient $x$ are represented as a sequence of visit records: $\mathbf{V} = \left[ \mathbf{v}^{(1)}, \mathbf{v}^{(2)}, \ldots, \mathbf{v}^{(N_x)} \right]$, where $\mathbf{v}^{(i)}$ corresponds to the $i$-th visit, and $N_x$ is the total number of visits for patient $x$. Each visit $\mathbf{v}^{(i)}$ contains diagnosis, procedure (e.g., surgery), and drug information, encoded as: $\mathbf{v}^{(i)} = \left[ \mathbf{v}_d^{(i)}, \mathbf{v}_p^{(i)}, \mathbf{v}_m^{(i)} \right]$, where $\mathbf{v}_d^{(i)} \in \{0,1\}^{|\mathcal{D}|}$, $\mathbf{v}_p^{(i)} \in \{0,1\}^{|\mathcal{P}|}$, and $\mathbf{v}_m^{(i)} \in \{0,1\}^{|\mathcal{M}|}$ are multi-hot encoded vectors representing diagnosis, procedure, and medication, respectively. And $\mathcal{D}, \mathcal{P}$, and $\mathcal{M}$ are corresponding code sets.

**Graph Representation.** Let $\mathcal{G} = (\mathbf{X}, \mathbf{A})$ denote a graph [6], where $\mathbf{X} \in \mathbb{R}^{N \times F}$ is the node feature matrix, with N and F denoting the number of nodes and feature dimensions, and $\mathbf{A} \in \mathbb{R}^{N \times N}$ is the adjacency matrix, with $\mathbf{A}_{ij} = 1$ if an edge exists between nodes $i$ and $j$, and otherwise, $\mathbf{A}_{ij} = 0$.

**Drug Combination Recommendation.** Given the longitudinal diagnosis sequence $\mathbf{v}_d^t = \left[ \mathbf{v}_d^{(1)}, \mathbf{v}_d^{(2)}, \ldots, \mathbf{v}_d^{(t)} \right]$ and procedure sequence : $\mathbf{v}_p^t = \left[ \mathbf{v}_p^{(1)}, \mathbf{v}_p^{(2)}, \ldots, \mathbf{v}_p^{(t)} \right]$, up to time $t$, as well as the DDI relation matrix $\mathbf{D}$, our objective is to learn a drug combination recommendation function $f(\cdot)$ that generates a multi-label output $\hat{y}^{(t)} \in \{0,1\}^{|\mathcal{M}|}$. Specifically, $\hat{y}^{(t)} = f(\mathbf{v}_d^t, \mathbf{v}_p^t)$.

## 2.2 Graph Information Bottleneck (GIB)

The Information Bottleneck (IB) principle [29] aims to extract a compact representation that retains maximal predictive information about the labels. Graph IB [36] extends this idea to irregular graph data, addressing the challenge of substructure recognition. Given a graph $\mathcal{G}$ and its label $\mathbf{Y}$, the optimal substructure $\mathcal{G}_{\text{IB}} = (\mathbf{X}_{\text{IB}}, \mathbf{A}_{\text{IB}})$ is obtained by optimizing the following objective:

$$\mathcal{G}_{\text{IB}} = \underset{\mathcal{G}_{\text{sub}}}{\arg\min} - I\left(\mathbf{Y}; \mathcal{G}_{\text{sub}}\right) + \beta I\left(\mathcal{G}; \mathcal{G}_{\text{sub}}\right), \tag{1}$$

where $\mathbf{X}_{\text{IB}}$ and $\mathbf{A}_{\text{IB}}$ denote the task-relevant feature set and the adjacency matrix of $\mathcal{G}$. $I(\cdot)$ is the mutual information term.

# 3 Methodology

As shown in Figure 2, **SubRec** consists of four components, which will be detailed below:

## 3.1 Patient Representation Module

We represent the patient's health status using diagnosis and procedure information, and define two learnable embedding matrices: the diagnosis embedding table $E_d \in \mathbb{R}^{|\mathcal{D}| \times F}$ and the procedure embedding table $E_p \in \mathbb{R}^{|\mathcal{P}| \times F}$ The embeddings are extracted via dot product between the multi-hot vectors and the embedding tables, and the resulting embeddings are summed as follows:

$$\mathbf{e}_d^{(i)} = \mathbf{v}_d^{(i)} \cdot E_d, \quad \mathbf{e}_p^{(i)} = \mathbf{v}_p^{(i)} \cdot E_p, \tag{2}$$

where $(\cdot)$ indicates matrix multiplication between two matrices. We use a single-layer Transformer Encoder [32] to model the historical diagnosis and procedure information, processing the entire sequence of embeddings up to the $i$-th visit. Given the sequences of diagnosis and procedure

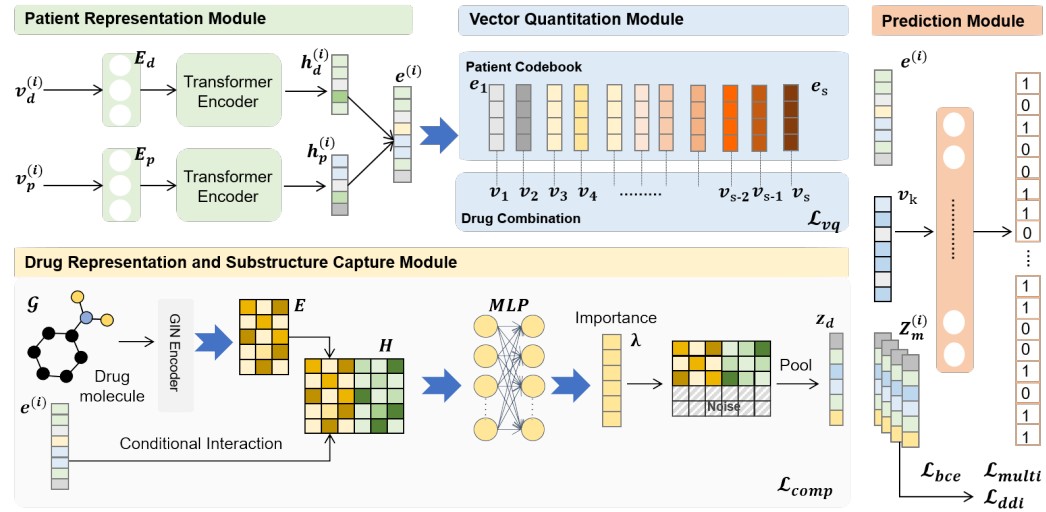

Figure 2: Overview of SubRec. First, diagnosis and procedure sequences are independently encoded by a Transformer-Encoder and concatenated to generate the patient representation $e^{(i)}$ at the $i$-th visit. Then, based on the GIB theory, drugs $d$ undergo core substructure extraction using $e^{(i)}$ as a conditional variable. The extracted core substructure vectors are pooled to form the drug substructure representation $z_d$, and collectively, these representations constitute $Z_m^{(i)}$. Simultaneously, $e^{(i)}$ is recorded as a codebook vector, and its corresponding drug recommendation result $v_k$ is stored. During the prediction phase, $e^{(i)}$, $Z_m^{(i)}$, and $v_k$ are utilized and combined to generate the final drug recommendation.

embeddings $\{\mathbf{e}_d^{(1)}, \mathbf{e}_d^{(2)}, \dots, \mathbf{e}_d^{(i)}\}$ and $\{\mathbf{e}_p^{(1)}, \mathbf{e}_p^{(2)}, \dots, \mathbf{e}_p^{(i)}\}$, the Transformer Encoder generates the corresponding output embeddings:

$$\mathbf{h}_d^{(i)} = \text{Encoder}_d(\{\mathbf{e}_d^{(1)}, \mathbf{e}_d^{(2)}, \dots, \mathbf{e}_d^{(i)}\}), \quad \mathbf{h}_p^{(i)} = \text{Encoder}_p(\{\mathbf{e}_p^{(1)}, \mathbf{e}_p^{(2)}, \dots, \mathbf{e}_p^{(i)}\}). \quad (3)$$

Next, we concatenate the diagnosis embedding $\mathbf{h}_d^{(i)}$ and procedure embedding $\mathbf{h}_p^{(i)}$ to form the final patient representation:

$$\mathbf{e}^{(i)} = \mathbf{h}_d^{(i)} \| \mathbf{h}_p^{(i)}. \quad (4)$$

### 3.2 Drug Representation and Substructure Capture Module

#### 3.2.1 Graph Encoding

We adopt GIN [38] to encode the molecular graph as $\mathbf{E} = \text{GIN}(\mathbf{X}, \mathbf{A})$, where $\mathbf{E} = \{\mathbf{E}_j\}_{j=1}^N$ denotes the node embeddings. To model interactions between nodes and a drug-level feature vector $\mathbf{e}^{(i)}$, we concatenate each $\mathbf{E}_j$ with $\mathbf{e}^{(i)}$ and feed the result into a multi-layer perceptron (MLP). The resulting interaction features are denoted as $\mathbf{H}$:

$$\mathbf{H} = \text{MLP}([\mathbf{E}; \mathbf{e}^{(i)}]), \quad (5)$$

where $[\mathbf{E}; \mathbf{e}^{(i)}]$ represents the concatenation of the $\mathbf{E} \in \mathbb{R}^{N \times F}$ and the vector $\mathbf{e}^{(i)} \in \mathbb{R}^F$ along the feature dimension, resulting in a matrix of size $\mathbb{R}^{N \times (2F)}$. The MLP operates independently on each row of the concatenated matrix, producing the interaction features $\mathbf{H} \in \mathbb{R}^{N \times F}$.

#### 3.2.2 Conditional Core Substructure Discovery

We focus on learning the core substructure $\mathcal{G}_{\text{CIB}} = (\mathbf{X}_{\text{CIB}}, \mathbf{A}_{\text{CIB}})$ of the input graph $\mathcal{G}$, conditioned on the paired patient representation $\mathbf{e}^{(i)}$.

**Definition 1.** *(Conditional Information Bottleneck, CIB)* Given random variables $V^1$, $V^2$, and $Y$, the CIB principle compresses $V^1$ into a bottleneck variable $T^1$, while preserving information relevant to predicting $Y$ conditioned on $V^2$. The objective is optimized as:

$$\min_{T^1} -I(Y; T^1 \mid V^2) + \beta I(X^1; T^1 \mid V^2), \quad (6)$$

where $\beta$ is a hyperparameter balancing trade-off between two conditional mutual information terms.

**Definition 2.** *(CIB-Graph)* Given a drug graph and patient representation $(\mathcal{G}, \mathbf{e}^{(i)})$ and their label information $\mathbf{Y}$, the optimal graph $\mathcal{G}_{\mathrm{CIB}} = (\mathbf{X}_{\mathrm{CIB}}, \mathbf{A}_{\mathrm{CIB}})$ discovered under the Conditional Information Bottleneck (CIB) principle is referred to as the CIB-Graph. It is defined as:

$$\mathcal{G}_{\mathrm{CIB}} = \arg\min_{\mathcal{G}_{\mathrm{sub}}} -I(\mathbf{Y}; \mathcal{G}_{\mathrm{sub}} \mid \mathbf{e}^{(i)}) + \beta I(\mathcal{G}; \mathcal{G}_{\mathrm{sub}} \mid \mathbf{e}^{(i)}), \tag{7}$$

By focusing on essential substructure information and reducing redundancy, $\mathcal{G}_{\mathrm{CIB}}$ offers a more efficient, task-specific way to address complexities of personalized medicine modeling.

***Minimizing*** $-I\left(\mathbf{Y}; \mathcal{G}_{sub} \mid \mathbf{e}^{(i)}\right)$. It ensures that the optimal substructure $\mathcal{G}_{\mathrm{sub}}$ retains sufficient information to predict $\mathbf{Y}$, conditioned on the patient representation $\mathbf{e}^{(i)}$. According to the chain rule of mutual information, the first term $-I\left(\mathbf{Y}; \mathcal{G}_{\mathrm{sub}} \mid \mathbf{e}^{(i)}\right)$ can be expanded as follows:

$$-I\left(\mathbf{Y}; \mathcal{G}_{\mathrm{sub}} \mid \mathbf{e}^{(i)}\right) = -I\left(\mathbf{Y}; \mathcal{G}_{\mathrm{sub}}, \mathbf{e}^{(i)}\right) + I\left(\mathbf{Y}; \mathbf{e}^{(i)}\right), \tag{8}$$

For the term $-I\left(\mathbf{Y}; \mathcal{G}_{\mathrm{sub}}, \mathbf{e}^{(i)}\right)$, introducing a variational approximation $p_\theta\left(\mathbf{Y} \mid \mathcal{G}_{\mathrm{sub}}, \mathbf{e}^{(i)}\right)$ for the intractable distribution $p\left(\mathbf{Y} \mid \mathcal{G}_{\mathrm{sub}}, \mathbf{e}^{(i)}\right)$, we can derive the following result based on the non-negativity property of the Kullback-Leibler (KL) divergence:

$$-I\left(\mathbf{Y}; \mathcal{G}_{\mathrm{sub}}, \mathbf{e}^{(i)}\right) = -\mathbb{E}_{\mathbf{Y}, \mathcal{G}_{\mathrm{sub}}, \mathbf{e}^{(i)}} \left[\log \frac{p\left(\mathbf{Y} \mid \mathcal{G}_{\mathrm{sub}}, \mathbf{e}^{(i)}\right)}{p(\mathbf{Y})}\right] \leq -\mathbb{E}_{\mathbf{Y}, \mathcal{G}_{\mathrm{sub}}, \mathbf{e}^{(i)}} \left[\log \frac{p_\theta\left(\mathbf{Y} \mid \mathcal{G}_{\mathrm{sub}}, \mathbf{e}^{(i)}\right)}{p(\mathbf{Y})}\right]$$

$$= -\mathbb{E}_{\mathbf{Y}, \mathcal{G}_{\mathrm{sub}}, \mathbf{e}^{(i)}} \left[\log p_\theta\left(\mathbf{Y} \mid \mathcal{G}_{\mathrm{sub}}, \mathbf{e}^{(i)}\right)\right] - H(\mathbf{Y}). \tag{9}$$

Here, $H(\mathbf{Y})$ represents the entropy of $\mathbf{Y}$ Consequently, the upper bound of $-I\left(\mathbf{Y}; \mathcal{G}_{\mathrm{sub}}, \mathbf{e}^{(i)}\right)$ is calculated by minimizing the prediction loss $\mathcal{L}_{\mathrm{pred}}\left(\mathbf{Y}, \mathcal{G}_{\mathrm{sub}}, \mathbf{e}^{(i)}\right)$.

$$\mathcal{L}_{\mathrm{pred}}\left(\mathbf{Y}, \mathcal{G}_{\mathrm{sub}}, \mathbf{e}^{(i)}\right) = \mathcal{L}(f(\mathbf{z}), \mathbf{Y}), \tag{10}$$

where $\mathbf{z} = \mathrm{pool}(\mathbf{H}_{\mathrm{sub}})$ is the graph-level embedding of $\mathcal{G}$, obtained from $\mathbf{H}_{\mathrm{sub}}$ of the subgraph $\mathcal{G}_{\mathrm{sub}}$, and $f$ denotes the prediction head. Details on extracting $\mathcal{G}_{\mathrm{sub}}$ are provided in Equation (13). The term $I\left(\mathbf{Y}; \mathbf{e}^{(i)}\right)$ is omitted, as minimizing it has been empirically shown to impair performance.

***Minimizing*** $I\left(\mathcal{G}; \mathcal{G}_{sub} \mid \mathbf{e}^{(i)}\right)$. We decompose it into:

$$I\left(\mathcal{G}; \mathcal{G}_{\mathrm{sub}} \mid \mathbf{e}^{(i)}\right) = I\left(\mathcal{G}_{\mathrm{sub}}; \mathcal{G}, \mathbf{e}^{(i)}\right) - I\left(\mathcal{G}_{\mathrm{sub}}; \mathbf{e}^{(i)}\right). \tag{11}$$

To address the inefficiency and instability in the optimization process caused by mutual information estimation, we introduce a method called noise injection. The core idea is to allow the model to inject noise into less informative substructures while introducing minimal noise into more informative ones. Specifically, given the node embedding $\mathbf{H}$, we compute a probability $p$ using a neural network $\mathbf{P}$:

$$p = \mathbf{P}(\mathbf{H}), \tag{12}$$

Based on the calculated $p$, we replace the node representation $\mathbf{H}$ with noise $\epsilon \sim N\left(\mu_\mathbf{H}, \sigma_\mathbf{H}^2\right)$ as:

$$\hat{\mathbf{H}} = \lambda \mathbf{H} + (1 - \lambda)\epsilon, \tag{13}$$

where $\lambda \sim \mathrm{Bernoulli}\left(\mathrm{Sigmoid}(p)\right)$, and $\mu_\mathbf{H}$ and $\sigma_\mathbf{H}^2$ are the mean and variance of $\mathbf{H}$, respectively.

To ensure the differentiability of the sampling process, we utilize the Gumbel-Sigmoid [20] function for the discrete random variable $\lambda$, defined as: $\lambda = \mathrm{Sigmoid}\left(\frac{1}{t}\log\left[\frac{p}{1-p}\right] + \log\left[\frac{u}{1-u}\right]\right)$, where $u \sim \mathrm{Uniform}(0, 1)$, and $t$ is the temperature parameter and 1.0 is chose. We minimize the upper bound of $I\left(\mathcal{G}_{\mathrm{sub}}; \mathcal{G}, \mathbf{e}^{(i)}\right)$ as follows:

$$I\left(\mathcal{G}_{\mathrm{sub}}; \mathcal{G}, \mathbf{e}^{(i)}\right) \leq \mathbb{E}_{\mathcal{G}, \mathbf{e}^{(i)}} \left[-\frac{1}{2}\log B + \frac{1}{2N^1}B + \frac{1}{2N^1}C^2\right],$$

$$-I\left(\mathcal{G}_{\mathrm{sub}}; \mathbf{e}^{(i)}\right) \leq \mathbb{E}_{\mathcal{G}_{\mathrm{sub}}, \mathbf{e}^{(i)}} \left[-\log p_\xi\left(\mathbf{e}^{(i)} \mid \mathcal{G}_{\mathrm{sub}}\right)\right], \tag{14}$$

where $B = \sum_{j=1}^{N^1} (1 - \lambda_j)^2$ and $C = \frac{\sum_{j=1}^{N^1} \lambda_j (\mathbf{H}_j - \mu_{\mathbf{H}})}{\sigma_{\mathbf{H}}}$, $p_\xi \left( \mathbf{e}^{(i)} \mid \mathcal{G}_{\text{sub}} \right)$ is the variational approximation of $p \left( \mathbf{e}^{(i)} \mid \mathcal{G}_{\text{sub}} \right)$. The final $\mathcal{L}_{\text{comp}}$ is optimized by refining the two variational upper bounds.

$$\mathbb{E}_{\mathcal{G}, \mathbf{e}^{(i)}} \left[ -\frac{1}{2} \log B + \frac{1}{2N} B + \frac{1}{2N} C^2 \right] + \mathbb{E}_{\mathcal{G}_{\text{sub}}, \mathbf{e}^{(i)}} \left[ -\log p_\xi \left( \mathbf{e}^{(i)} \mid \mathcal{G}_{\text{sub}} \right) \right] := \mathcal{L}_{\text{comp}} \qquad (15)$$

Each drug $d$ undergoes the above substructure extraction process with $\mathbf{e}^{(i)}$ to obtain $z_d$, Finally, all representations of drug molecules are collected together and compose an embedding table $Z_m^{(i)} \in \mathbb{R}^{|M| \times F}$, of which each row corresponds to a drug.

### 3.3 Vector Quantitation Module (VQ)

In this section, we introduce the VQ to enhance drug recommendations by utilizing all patients information from the database. Specifically, for each visit $i$, we do not only record the prior visit information of the current patient, but also store all visits from other patients up to the current patient as auxiliary information, denoted as $e^{(i)} \to v_m^{(i)}$, where $e^{(i)}$ represents the health representation of patient $x$ at visit $i$, and $v_m^{(i)}$ is the multi-hot encoded drug information at the corresponding visit.

However, directly storing all this information would result in excessive memory and time consumption, as the dataset can grow very large. To address this issue, we introduce the concept of vector quantization [30] to create a trainable, discrete codebook, which helps to compress the vast medical history into a fixed number of prototypes. The codebook $W$ is defined as follows:

$$W = \{e_1 : v_1, e_2 : v_2, \ldots, e_S : v_S\}, \qquad (16)$$

where each pair $(e_m, v_m)$ corresponds to a discrete entry, with $e_m$ being a patient's health representation and $v_m$ being the associated drug recommendation. The parameter $S$ represents the number of distinct patient-drug combinations in the latent space.

For each test patient visit, we search for the most similar health representation in this latent space by using a nearest-neighbor approach, indexed by $k$. This process serves as a non-linearity that maps the latent vector $e^{(i)}$ to one of the $M$ codebook entries. The selection process is as follows:

$$q(k \mid e^{(i)}) = \begin{cases} 1, & \text{if } k = \arg\min_j \|e^{(i)} - e_j\|_2^2, \\ 0, & \text{otherwise.} \end{cases} \qquad (17)$$

Once index $k$ is identified, the corresponding drug recommendation $v_k$ is retrieved from the codebook.

To update the codebook and ensure that the encoder's output remains close to the selected codebook embedding, we define the vector quantization loss $\mathcal{L}_{\text{vq}}$ as follows, where $\text{sg}[\cdot]$ denotes the stop-gradient operation and $\delta$ is a hyperparameter. This loss function consists of four terms:

$$\mathcal{L}_{\text{vq}} = \|\text{sg}[e^{(i)}] - e_k\|_2^2 + \delta\|e^{(i)} - \text{sg}[e_k]\|_2^2 + \|\text{sg}[v_m^{(i)}] - v_k\|_2^2 + \delta\|v_m^{(i)} - \text{sg}[v_k]\|_2^2. \qquad (18)$$

Here, $v_m^{(i)}$ represents the actual drug prescribed to the patient at visit $i$, and $v_k$ is the drug associated with the selected codebook entry $e_k$. The last two terms in the loss function ensure the selected drug vector $v_k$ is close to the actual prescribed drug $v_m^{(i)}$ for accurate drug recommendations.

As $\mathcal{L}_{\text{vq}}$ gradually converge, we obtain a stable codebook $W$, which clusters the infinite possible combinations of conditions and medications into a discretized set of $M$ finite pairs, enabling efficient and context-aware drug recommendations for new patients. The well-trained codebook provides insights into potential patient-drug distributions.

### 3.4 Recommendation Prediction Module

In this module, we leverage the outputs from the previous three modules: the patient representation $\mathbf{e}^{(i)}$, the drug molecular representation $Z_m$, and the recommended drug embedding $v_k$ retrieved via codebook. Using these components, we apply attention based reading procedure calculate the output $o_m^{(i)}$ to retrieve the most relevant information with respect to the query $\mathbf{e}^{(i)}$.

$$o_m^{(i)} = \text{Softmax}(\mathbf{e}^{(i)} \cdot \left( Z_m^{(i)} \right)^T) \cdot Z_m^{(i)} \qquad (19)$$

Additionally, we can obtain prior drug category information from the codebook. To further refine the drug recommendation, we calculate the output $o_d^{(i)}$ as follows:

$$o_d^{(i)} = v_k \cdot Z_m^{(i)}. \tag{20}$$

The final step to utilize patient representation and memory output to predict the multi-label drug:

$$\hat{y}^{(i)} = \sigma\left(\left[e^{(i)}, o_m^{(i)}, o_d^{(i)}\right]\right), \tag{21}$$

where $\sigma$ is the sigmoid function, and $\hat{y}^{(i)}$ represents appearance probability of each drug in the prescription. Then we can obtain a multi-hot prediction vector $\hat{\mathbf{o}}^{(i)}$ by picking out the entries of $\hat{y}^{(i)}$ whose value is greater than a predefined threshold value $\tau$ (in this work, $\tau$ is 0.5).

### 3.5 Loss function

In this section, we summarize our loss function. We instantiate $\mathcal{L}_{\text{pred}}$ Equation (10), which includes both binary cross-entropy loss $\mathcal{L}_{\text{bce}}$ and multi-label margin loss $\mathcal{L}_{\text{multi}}$. Specifically, $\mathcal{L}_{\text{bce}}$ is the binary cross-entropy loss, while $\mathcal{L}_{\text{multi}}$ is the multi-label margin loss, which ensures that the predicted probability for ground truth labels is at least 1 margin larger than for other labels.

$$\mathcal{L}_{\text{bce}} = -\sum_{j=1}^{|\mathcal{M}|}\left[o_j^{(i)}\log(\hat{y}_j^{(i)}) + (1 - o_j^{(i)})\log(1 - \hat{y}_j^{(i)})\right],$$

$$\mathcal{L}_{\text{multi}} = \sum_{p,q:o_p^{(i)}=1,o_q^{(i)}=0} \frac{\max(0, 1 - (\hat{y}_p^{(i)} - \hat{y}_q^{(i)}))}{|\mathcal{M}|}, \tag{22}$$

$$\mathcal{L}_{\text{pred}} = \theta \cdot \mathcal{L}_{\text{bce}} + (1 - \theta) \cdot \mathcal{L}_{\text{multi}}.$$

where the superscript $i$ denotes the $i$-th entry of the vector.

**DDI Loss.** We define the DDI loss as [40]:

$$\mathcal{L}_{\text{DDI}} = \sum_{i}\sum_{p=1}^{|M|}\sum_{q=1}^{|M|}(\hat{y}_p^{(i)} \cdot \hat{y}_q^{(i)}) \cdot D_{pq}, \tag{23}$$

where $(\hat{y}_p^{(i)} \cdot \hat{y}_q^{(i)}) \cdot D_{pq}$ gives the pairwise DDI probability. Subsequently, the final total loss is:

$$\mathcal{L} = \alpha \cdot (\mathcal{L}_{\text{pred}} + \beta\mathcal{L}_{\text{comp}} + \gamma\mathcal{L}_{\text{vq}}) + (1 - \alpha) \cdot \mathcal{L}_{\text{DDI}}, \tag{24}$$

where $\alpha$, $\beta$, $\theta$, and $\gamma$ are hyperparameters that control the trade-offs between different losses. $\mathcal{L}_{\text{pred}}$ and $\mathcal{L}_{\text{comp}}$ compress $\mathcal{G}$ into a substructure $\mathcal{G}_{\text{sub}}$ while retaining the minimal information relevant to the task, with $\beta$ controlling the balance between prediction and compression. $\mathcal{L}_{\text{vq}}$ updates the environment codebook, with $\gamma$ governing the update process. To control DDI rates, we dynamically adjust $\alpha$ during training to balance prediction and safety [40].

## 4 EXPERIMENTS

In this section, we conduct extensive experiments to answer the following questions:
- **RQ1:** Can SubRec improve the accuracy of combinatorial drug recommendations?
- **RQ2:** To what extent do the proposed VQ and GIB modules enhance the performance?
- **RQ3:** What do the codebook vectors represent and learn within the model?

### 4.1 Experimental Settings

Here, we briefly introduce the dataset, baseline models, evaluation metrics, and configurations.

**Dataset.** We use the EHR data from **MIMIC-III** and **MIMIC-IV** [13]. These dataset include various patients and clinical events. More detailed data analysis results can be found in Appendix B.

Table 1: Performance of different methods on MIMIC-III and MIMIC-IV. (Underlined are the best baseline results, while the top-performing method is highlighted in bold which is under t-tests, at the 95% confidence level. The calculation method of DDI rate is consistent with previous studies [27, 40, 35].)

| Method | MIMIC-III | | | | | MIMIC-IV | | | | |
|---|---|---|---|---|---|---|---|---|---|---|
| | Jaccard (↑) | PRAUC (↑) | F1 (↑) | DDI (↓) | #MED | Jaccard (↑) | PRAUC (↑) | F1 (↑) | DDI (↓) | #MED |
| **General Methods** | | | | | | | | | | |
| ECC | 0.4935 (0.0021) | 0.7634 (0.0020) | 0.6512 (0.0018) | 0.0788 (0.0009) | 16.2579 (0.0982) | 0.4152 (0.0023) | 0.6783 (0.0022) | 0.5651 (0.0019) | 0.0732 (0.0013) | 9.8724 (0.2078) |
| LEAP | 0.4521 (0.0043) | 0.6581 (0.0029) | 0.6152 (0.0037) | 0.0720 (0.0015) | 18.6742 (0.0658) | 0.3909 (0.0039) | 0.5542 (0.0014) | 0.5439 (0.0023) | 0.0550 (0.0014) | 12.6265 (0.1263) |
| 4SDrug | 0.5210 (0.0025) | 0.7780 (0.0028) | 0.6762 (0.0021) | 0.0781 (0.0007) | 16.1684 (0.0982) | 0.4401 (0.0023) | 0.6833 (0.0018) | 0.5933 (0.0016) | 0.0718 (0.0006) | 12.8924 (0.2043) |
| **Longitudinal History Considered** | | | | | | | | | | |
| COGNet | 0.5231 (0.0019) | 0.7608 (0.0012) | 0.6676 (0.0014) | 0.0737 (0.0007) | 28.7450 (0.1152) | 0.4483 (0.0001) | 0.6512 (0.0015) | 0.5950 (0.0013) | 0.0866 (0.0009) | 15.4624 (0.1245) |
| RETAIN | 0.4868 (0.0034) | 0.6472 (0.0032) | 0.6523 (0.0026) | 0.0759 (0.0035) | 18.5941 (0.2186) | 0.4387 (0.0013) | 0.4518 (0.0027) | 0.5023 (0.0025) | 0.0825 (0.0025) | 15.9743 (0.1537) |
| VITA | 0.5412 (0.0018) | 0.7720 (0.0012) | 0.6838 (0.0005) | 0.0630 (0.0006) | 19.5941 (0.1929) | 0.4420 (0.0016) | 0.6995 (0.0011) | 0.6002 (0.0014) | 0.0510 (0.0006) | 12.5123 (0.1527) |
| GAMENet | 0.5119 (0.0029) | 0.5190 (0.0023) | 0.6676 (0.0027) | **0.0610** (0.0009) | 20.9423 (0.1646) | 0.4495 (0.0031) | 0.4353 (0.0034) | 0.6033 (0.0023) | **0.0502** (0.0007) | 14.5024 (0.1542) |
| **Substructure-Based Methods** | | | | | | | | | | |
| SafeDrug | 0.5167 (0.0030) | 0.7681 (0.0028) | 0.6724 (0.0027) | 0.0628 (0.0005) | 20.2601 (0.1079) | 0.4483 (0.0033) | 0.6858 (0.0030) | 0.6098 (0.0029) | 0.0609 (0.0007) | 14.0723 (0.1064) |
| MoleRec | 0.5303 (0.0032) | 0.7795 (0.0030) | 0.6844 (0.0026) | 0.0692 (0.0008) | 21.0893 (0.1788) | 0.4580 (0.0035) | 0.6867 (0.0031) | 0.6040 (0.0033) | 0.0699 (0.0007) | 14.0525 (0.1583) |
| **Our Method** | | | | | | | | | | |
| **SubRec** | **0.5585** (0.0035) | **0.7927** (0.0025) | **0.7016** (0.0030) | 0.0623 (0.0008) | 19.5040 (0.1357) | **0.4635** (0.0025) | **0.7023** (0.0014) | **0.6216** (0.0025) | 0.0674 (0.0005) | 14.0634 (0.1357) |

**Baselines.** In our comprehensive evaluation, our model is compared with nine SOTA baselines, grouped into three categories. The first includes general methods based on patient representations, including ECC [24], LEAP [43], and 4SDrug [27]. The second category includes experience/rule-based methods that leverage longitudinal patient history, such as RETAIN [2], VITA [16], GAMENet [25], and COGNet [35]. The third category consists of drug recommendation methods that consider molecular substructures, including SafeDrug [39] and MoleRec [40]. Details are in Appendix C.

**Evaluation Metrics.** Four metrics are employed to evaluate model performance: DDI Rate, Jaccard Similarity Score (Jaccard), F1-score, and Precision-Recall AUC (PRAUC). The Jaccard Similarity Score measures the similarity between predicted and true drug sets, while the F1-score and PRAUC are used to assess the model's classification performance. The DDI Rate focuses on controlling the occurrence of adverse drug interactions [12, 10, 47, 1]. The detailed definitions are in Appendix D.

## 4.2 Model performance

Similar to previous studies [25, 40], the longitudinal patient history is split into training, validation, and test sets with a ratio of $4 : 1 : 1$ for evaluating model performance. The experimental results are recorded in Table 1 and Appendix F. Based on these outcomes, we delineate three key observations:

**Obs.1: The drug recommendation results exhibit higher similarity when patient cases are similar**. We analyzed the MIMIC-III dataset, using a patient's visit record as the reference point. As shown in Appendix F.1, we randomly selected 10,000 samples and calculated the diagnosis and drug similarity using the Jaccard index as the similarity measure. The Pearson correlation coefficient between the two was found to be 0.53, indicating a relatively positive correlation between the patient visit characteristics and the recommended drug combinations. This result is consistent with clinical understanding, as similar diagnoses often lead to the prescription of similar drugs.

**Obs.2: SubRec exhibits optimal predictive performance compared to other baseline models**. Specifically, ECC and LEAP perform relatively poorly as they only consider diagnoses and procedures from the current visit. In contrast, RETAIN, VITA, and GAMENet perform better because they take longitudinal patient information into account. SafeDrug and MoleRec incorporate drug molecule structures in drug recommendation, leading to further performance improvements. However, these models, which adopt the BRICS method [3], focus solely on molecular features and ignore the relationships between patient health conditions and molecular substructures. Consequently, SubRec outperforms these two models by integrating both aspects.

**Obs.3: SubRec exhibits better robustness in cases with limited or no historical drug information.** The MIMIC-III dataset has an imbalanced distribution of patient visit records, with the majority of patients visiting the hospital fewer than five times. This creates a challenge for models that need to learn patient-drug relationships from other patients' visit records for accurate recommendations. As shown in Appendix F.2, overall, models tend to perform better with more visits, suggesting that historical visit records contribute to the model's inference process. However, SubRec demonstrates superior prediction performance in this scenario while maintaining a low and stability DDI rate.

### 4.3 Sensitivity analysis

In this section, we investigate the impact of $\beta$, $\gamma$, $\delta$ and codebook size $S$ on model performance.

**Obs.4: The parameter $\beta$ aims to achieve optimal prediction performance while maintaining a reasonable substructure compression rate.** We first analyze the effect of the parameter $\beta$ in the substructure extraction process, which controls the trade-off between prediction accuracy and compression efficiency in our final objective, as outlined in Equation (7). When $\beta = 0$, the model performs poorly, as it preserves the original graph input without capturing the essential substructure information. As $\beta$ increases, the model's performance improves, indicating that some compression is beneficial for prediction. However, excessively large values of $\beta$ do not result in continuous performance gains, as overly aggressive compression can distort the substructure and hinder the model's ability to make accurate predictions. For example, when $\beta > $ 5E-4, a slight decline in performance is observed. Therefore, $\beta = $ 5E-4 is selected.

**Obs.5: We also conducted a sensitivity analysis on the hyperparameters $\delta$ and $\gamma$ within the loss function $\mathcal{L}_{\mathbf{vq}}$.** The hyperparameter $\delta$ is used to balance the commitment loss and embedding loss, while $\gamma$ represents the weight of $\mathcal{L}_{\mathrm{vq}}$ in the total loss function, as outlined in Equation (18) and Equation (24). We tested multiple combinations of these hyperparameters, and observed that varying $\delta$ and $\gamma$ did not significantly affect the model's performance, with overall performance remaining stable. This finding is consistent with the other studies [30, 34, 46]. Therefore we set $\delta$ to 0.40 and $\gamma$ to 5E-4 (in Appendix G).

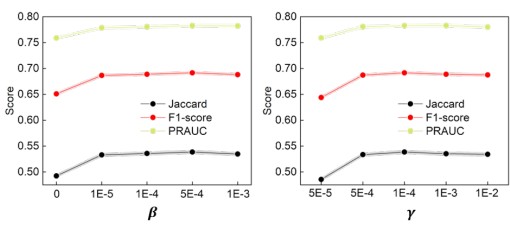

Figure 3: Sensitivity analysis results for $\beta$ and $\gamma$.

**Obs.6: The codebook size has a slight impact on the recommendation results but significantly influences the training duration.** We conducted an in-depth investigation into the effect of varying the number of vector embeddings on our model's performance, as shown in Figure 5. The results reveal that changes in the number of codebook embeddings have a negligible impact on test performance across test datasets, demonstrating the robustness of our model. On the other hand, increasing the number of codebook vectors significantly increases training time and memory consumption (more in Appendix G). To balance these factors, we chose 32 as the final configuration for our model.

### 4.4 Ablation study

To verify the effectiveness of each module of SubRec, we design the following variants:

- *Replace the Transformer module with an RNN module in the Patient encoder.*
- *w/o substructure capture. SubRec no longer considers $Z_m^{(i)}$ when assessing substructure impact.*
- *Remove the auxiliary codebook. Replace codebook vectors with noise.*

**Obs.7: Ablation on model components.** As shown in Table 2, when the RNN module is used as a replacement for the Transformer, the model performance slightly decreases. This may be due to RNN is less effective at capturing long-range dependencies in EHRs compared to Transformer. Removing either the substructure capture or auxiliary codebook results in a noticeable performance drop. This suggests that SubRec effectively models the relevancy between patients and molecular substructures, thereby enhancing the drug recommendation process. The role of the auxiliary codebook is more prominent in enabling the model to learn and summarize knowledge from historical patient records. Without this knowledge, the recommendation results deteriorate obviously.

### 4.5 Case study

**Obs.8: Different embeddings in the codebook exhibit clear boundaries in the visualization.** This indicates that the model effectively captures a diverse range of patient-drug variables. Moreover, embeddings derived from each distinct environment form tight clusters around their corresponding environment embeddings. This suggests that updating the codebook vectors essentially clusters the molecular embeddings, with the environment embeddings acting as cluster centers (in Appendix H).

Table 2: Ablation study on MIMIC-III in terms of DDI rate, Jaccard, F1-score, and PRAUC.

| Method | DDI ($\downarrow$) | Jaccard ($\uparrow$) | F1-score ($\uparrow$) | PRAUC ($\uparrow$) | Avg.# of Drugs |
|---|---|---|---|---|---|
| Patient Encoder (Transformer$\rightarrow$RNN) | $0.0737_{(0.0009)}$ | $0.5353_{(0.0031)}$ | $0.6886_{(0.0027)}$ | $0.7812_{(0.0026)}$ | $19.6349_{(0.1676)}$ |
| w/o substructure capture | $0.0748_{(0.0007)}$ | $0.5228_{(0.0041)}$ | $0.6752_{(0.0034)}$ | $0.7701_{(0.0026)}$ | $19.3808_{(0.1599)}$ |
| w/o auxiliary codebook | $0.0729_{(0.0005)}$ | $0.5257_{(0.0031)}$ | $0.6810_{(0.0026)}$ | $0.7723_{(0.0026)}$ | $19.2589_{(0.1646)}$ |
| SubRec | $\mathbf{0.0623}_{(0.0008)}$ | $\mathbf{0.5585}_{(0.0035)}$ | $\mathbf{0.7016}_{(0.0030)}$ | $\mathbf{0.7827}_{(0.0025)}$ | $19.5040_{(0.1357)}$ |

## 5 Conclusion

In this work, we propose **SubRec**, a novel framework for personalized drug recommendation. SubRec first encodes historical patient visit records and quantize to to construct a discrete codebook, while simultaneously capturing the complex relationships between patient health conditions and drug substructures. The quantization process eliminates the need to model variational distributions in high-dimensional latent spaces, resulting in a more stable and lightweight training process. By integrating substructure representations and discrete prototypes, SubRec improves training stability and efficiency, while enabling accurate and clinically reliable recommendations on real-world datasets.

## 6 Acknowledgement

This paper is partially supported by the National Natural Science Foundation of China (No.12227901). The AI-driven experiments, simulations and model training were performed on the robotic AI-Scientist platform of Chinese Academy of Sciences., Anhui Science Foundation for Distinguished Young Scholars (No.1908085J24), Natural Science Foundation of China (No.62502491).

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

# A    Related Work

In this work, we classify drug recommendation research into two main categories: experience/rule-based drug recommendation models and drug molecular structure-based approaches.

**Experience/rule-based methods** learn drug recommendation paradigms from historical data. These methods leverage current patient health conditions or longitudinal patient history, summarizing experiential knowledge to perform drug combination recommendations. LEAP [43] formulates drug recommendation as a multi-instance multi-label (MIML) task and utilizes attention mechanisms to capture dependencies between drug labels during a patient's current visit to generate prescriptions. 4SDrug [27] proposes measuring the similarity between symptom and drugsets for recommendation. **Researchers have increasingly realized that incorporating longitudinal patient history leads to more effective recommendations**. Consequently, RETAIN [2] develops a two-level neural attention model for healthcare time-series prediction. GameNet [25] utilizes memory-augmented neural networks, storing a patient's longitudinal visit history as references for future predictions. ALGNET [22] further enhances GameNet by incorporating Dual-Self-Attention and Light-GCN to address challenges related to the lack of long-range dependency information. COGNet [35] introduces a novel copy-or-predict mechanism to achieve drug recommendation. While these methods perform successfully, they overlook the core functional substructures on drug selection.

**Drug Structure-considered methods** emphasize the relationship between the drug molecular substructure and a patient's health condition to enhance drug recommendations. For example, Safe-Drug [39] proposes a DDI-controllable drug recommendation model that leverages molecular structures and more effectively models drug-drug interactions (DDIs). MoleRec [40] adopts the BRICS method [3] to decompose drug molecules into substructures and models the relationships between patient health conditions and molecular substructures to improve prediction accuracy. However, despite the advantages of molecular substructure-aware models in improving efficacy and explainability, neglecting patient health conditions when extracting substructures may disrupt the integrity of functional groups within drug molecules. This could lead to misguided model predictions regarding the relevance between a given patient's health condition and the corresponding molecular substructures.

**Graph Information Bottleneck (GIB)** theory provides a precise methodology for extracting subgraphs and has been widely applied in the task of subgraph extraction from a single graph. PGIB [42] introduces a GIB framework designed to identify informative yet compact subgraphs from the original graph, addressing key graph learning challenges such as graph denoising and compression. To optimize the challenging GIB objective, PGIB incorporates a mutual information estimator tailored for irregular graph data, a bi-level optimization scheme, and a connectivity loss to stabilize the optimization process. VGIB [41] further stabilizes the subgraph extraction process by introducing Gaussian noise into node representations, modulating the information flow from the original graph to the perturbed graph. Here, directly applying the GIB theory to extract core substructures proves challenging, as the importance of a substructure is context-dependent, i.e., patient's health condition, which requires further consideration.

# B    The statistics result of dataset

The statistics are summarized in Table 3. Specifically, we record the total number of patients, clinical events, diagnoses, procedures, and medications, along with the average number of visits per patient, average diagnoses per visit, and other related information.

# C    Baselines

A detailed description of the baselines is presented here:

**Ensemble Classifier Chain (ECC)** [24]: This multi-label model organizes logistic regression (LR) classifiers into a sequential chain, where each classifier receives the predictions from the preceding classifier as additional features.

**LEAP** [43]: LEAP formulates drug recommendation as a sequential decision-making process, employing a recurrent decoder to model label dependencies and a content-based attention mechanism to capture the mapping between labels and instances.

Table 3: Statistics of processed data.

| Item | MIMIC-III | MIMIC-IV |
|---|---|---|
| # Visits / Patients | 14949 / 6344 | 19461 / 7567 |
| Disease / Proc. Size | 1959 / 1440 | 3973 / 1338 |
| ATC3 / ATC4 Size | 112 / 141 | 212 / 302 |
| Avg. / Max Visits | 4.92 / 29 | 7.28 / 42 |
| Avg. / Max Diag. | 13.79 / 39 | 13.39 / 39 |
| Avg. / Max Proc. | 4.40 / 28 | 2.57 / 28 |
| Avg. / Max ATC3 | 19.58 / 52 | 10.82 / 55 |
| Avg. / Max ATC4 | 26.23 / 63 | 13.31 / 70 |

**4SDrug** [27]: 4SDrug is designed to recommend small sets of drugs, aiming to minimize drug-drug interactions (DDIs) while ensuring effective treatment options.

**RETAIN** [2]: RETAIN utilizes a two-level neural attention mechanism that identifies influential past visits and significant clinical variables within those visits, enabling better interpretation of temporal healthcare data.

**VITA** [16]: VITA aims to recommend effective medications for patients' current visits by leveraging information from their present and past medical histories. It identifies relevant historical medical visits for each patient's current condition and accurately quantifies the correlation between the current visit and each historical visit.

**GAMENet** [25]: GAMENet is based on memory networks enhanced with a memory bank that integrates drug usage information, Drug-Drug Interaction (DDI) graphs, and dynamic memory to incorporate patient history into predictions.

**SafeDrug** [39]: SafeDrug extracts and encodes molecular structure information to enrich the drug recommendation process, improving drug selection by considering structural characteristics.

**MoleRec** [40]: investigates the relationships between the health condition of patients and molecular substructures to improve the prediction.

**COGNet** [35]: COGNet proposes a novel approach where it decides whether to replicate a previously prescribed drug or recommend a new drug combination by analyzing historical recommendations and the current patient visit.

## D    Evaluation Metrics calculation method

Here, we provide the detailed calculation methods for the four metrics used to evaluate model performance: Drug-Drug Interaction Rate (DDI), Jaccard Similarity Score (Jaccard), F1-score, and Precision-Recall AUC (PRAUC).

**Drug-Drug-Interaction Rate (DDI)** For a certain patient $x$, the corresponding DDI is defined as:

$$DDI = \frac{\sum_{i=1}^{N_x} \sum_{k,l \in \{j:\hat{o}_j^{(t)}=1\}} 1\{D_{kl}=1\}}{\sum_{i=1}^{N_x} \sum_{k,l \in \{j:\hat{o}_j^{(t)}=1\}} 1}, \tag{25}$$

where $N_x$ represents the total number of visits for patient $x$, $o^{(t)}$ denotes the multi-label predictions at the $t$-th visit, $o_j^{(t)}$ denotes the $j$-th entry of $o^{(t)}$, $D$ is the prior DDI relation matrix and $1$ is an indicator function which returns 1 when $D_{kl}=1$, otherwise 0.

**Jaccard Similarity Score (Jaccard)** For a certain patient $x$ at the $t$-th visit, the definition of Jaccard is as follows:

$$\text{Jaccard}^{(t)} = \frac{\left| \left\{ i : \hat{o}_i^{(t)} = 1 \right\} \cap \left\{ i : o_i^{(t)} = 1 \right\} \right|}{\left| \left\{ i : \hat{o}_i^{(t)} = 1 \right\} \cup \left\{ i : o_i^{(t)} = 1 \right\} \right|}, \tag{26}$$

where $\hat{o}^{(t)}$ and $o^{(t)}$ denote the multi-label predictions and ground-truth recommendation, respectively. Note that $*_i$ represents the i-th entry of $*$. Then, we take the average over all the patient's visits to obtain the final Jaccard Similarity Score for patient $x$,

$$\text{Jaccard} = \frac{1}{N_x} \sum_{i=1}^{N_x} \text{Jaccard}^{(i)}, \tag{27}$$

where $N_x$ represents the total number of visits for patient $x$.

**F1-score** We first provide the definitions of Precision and Recall for a patient $x$ at the $t$-th visit,

$$\text{Precision}^{(t)} = \frac{|\{i : \hat{o}_i^{(t)} = 1\} \cap \{i : o_i^{(t)} = 1\}|}{|\{i : \hat{o}_i^{(t)} = 1\}|}, \tag{28}$$

$$\text{Recall}^{(t)} = \frac{|\{i : \hat{o}_i^{(t)} = 1\} \cap \{i : o_i^{(t)} = 1\}|}{|\{i : o_i^{(t)} = 1\}|}. \tag{29}$$

The F1-score is the harmonic mean of Precision and Recall,

$$F1^{(t)} = \frac{2}{\frac{1}{\text{Precision}^{(t)}} + \frac{1}{\text{Recall}^{(t)}}}. \tag{30}$$

Then, we average over all visits and obtain F1 score for patient $x$,

$$F1 = \frac{1}{N_x} \sum_{i=1}^{N_x} F1^{(i)}, \tag{31}$$

where $N_x$ represents the total number of visits for patient $x$.

**Precision Recall AUC (PRAUC)** Note that we treat drug combination recommendation as an information retrieval problem. For the patient $x$ at the $t$-th visit, PRAUC is defined as follows:

$$\text{PRAUC}^{(t)} = \sum_{k=1}^{|M|} \text{Precision}(k)^{(t)} \Delta \text{Recall}(k)^{(t)}, \tag{32}$$

$$\Delta \text{Recall}(k)^{(t)} = \text{Recall}(k)^{(t)} - \text{Recall}(k-1)^{(t)}, \tag{33}$$

where $k$ is the rank in the sequence of the retrieved drugs, $|M|$ denotes the number of drugs. $\text{Precision}(k)^{(t)}$ represents the precision at cut-off $k$ in the ordered retrieval list and $\text{Recall}(k)^{(t)}$ denotes the change of recall from drug $k-1$ to $k$. We also average over all visits and then obtain the PRAUC value for patient $x$.

$$\text{PRAUC} = \frac{1}{N_x} \sum_{i=1}^{N_x} \text{PRAUC}^{(i)}, \tag{34}$$

where $N_x$ represents the total number of visits for patient $x$.

# E   Proof

The proof of Equation (9) is as follows

$$-I(Y; \mathcal{G}_{sub}, e^{(i)}) = -\mathbb{E}_{Y, \mathcal{G}_{sub}, e^{(i)}} \left[ \log \frac{p(Y \mid \mathcal{G}_{sub}, e^{(i)})}{p(Y)} \right] \tag{35}$$

Since the true conditional distribution $p(Y \mid X)$ is intractable, we introduce a variational approximation $p_\theta(Y \mid X)$. Then:

$$-\mathbb{E}_{Y,\mathcal{G}_{sub},e^{(i)}} \left[ \log \frac{p(Y \mid \mathcal{G}_{sub}, e^{(i)})}{p(Y)} \right]$$

$$= -\mathbb{E}_{Y,\mathcal{G}_{sub},e^{(i)}} \left[ \log \frac{p_\theta(Y \mid \mathcal{G}_{sub}, e^{(i)})}{p(Y)} \cdot \frac{p(Y \mid \mathcal{G}_{sub}, e^{(i)})}{p_\theta(Y \mid \mathcal{G}_{sub}, e^{(i)})} \right] \tag{36}$$

$$= -\mathbb{E}_{Y,\mathcal{G}_{sub},e^{(i)}} \left[ \log \frac{p_\theta(Y \mid \mathcal{G}_{sub}, e^{(i)})}{p(Y)} \right] - \mathbb{E}_{Y,\mathcal{G}_{sub},e^{(i)}} \left[ \log \frac{p(Y \mid \mathcal{G}_{sub}, e^{(i)})}{p_\theta(Y \mid \mathcal{G}_{sub}, e^{(i)})} \right] \tag{37}$$

$$= -\mathbb{E}_{Y,\mathcal{G}_{sub},e^{(i)}} \left[ \log \frac{p_\theta(Y \mid \mathcal{G}_{sub}, e^{(i)})}{p(Y)} \right] - \mathbb{E}_{\mathcal{G}_{sub},e^{(i)}} \left[ D_{KL} \left( p(Y \mid \mathcal{G}_{sub}, e^{(i)}) \| p_\theta(Y \mid \mathcal{G}_{sub}, e^{(i)}) \right) \right] \tag{38}$$

By the non-negativity of the KL divergence, we have:

$$-\mathbb{E}_{Y,\mathcal{G}_{sub},e^{(i)}} \left[ \log \frac{p(Y \mid \mathcal{G}_{sub}, e^{(i)})}{p(Y)} \right] \leq -\mathbb{E}_{Y,\mathcal{G}_{sub},e^{(i)}} \left[ \log \frac{p_\theta(Y \mid \mathcal{G}_{sub}, e^{(i)})}{p(Y)} \right] \tag{39}$$

Continuing, we can rewrite the right-hand side as:

$$-\mathbb{E}_{Y,\mathcal{G}_{sub},e^{(i)}} \left[ \log \frac{p_\theta(Y \mid \mathcal{G}_{sub}, e^{(i)})}{p(Y)} \right] = -\mathbb{E}_{Y,\mathcal{G}_{sub},e^{(i)}} \left[ \log p_\theta(Y \mid \mathcal{G}_{sub}, e^{(i)}) \right] + \mathbb{E}_Y [\log p(Y)] \tag{40}$$

$$= -H(Y) - \mathbb{E}_{Y,\mathcal{G}_{sub},e^{(i)}} \left[ \log p_\theta(Y \mid \mathcal{G}_{sub}, e^{(i)}) \right] \tag{41}$$

where $H(Y) = -\mathbb{E}_Y [\log p(Y)]$ is the entropy of $Y$. $-\mathbb{E}_{Y,\mathcal{G}_{sub},e^{(i)}} \left[ \log p_\theta(Y \mid \mathcal{G}_{sub}, e^{(i)}) \right]$ is calculated by the prediction loss in Eq. 10, $\mathcal{L}_{\text{pred}}(Y, \mathcal{G}_{sub}, e^{(i)})$, where $\mathbf{z} = \text{pool}(\mathbf{H}_{sub})$, the graph-level embedding of $\mathcal{G}$ obtained from $\mathbf{H}_{sub}$ of the subgraph $\mathcal{G}_{sub}$, and $f$ denotes the prediction head. $H(Y)$ is a constant value that can be ignored.

## F  Additional Experimental Results

### F.1  Configurations

The hidden size is set to 128. The model is trained by Adam optimizer [5]. The code is based on Python 3.8.16 and PyTorch 1.9.0. The model was trained on an NVIDIA Tesla V100 16GB. We applied bootstrapping sampling 10 times on the validation set, and the results are presented as the mean and standard deviation.

### F.2  Similarity Analysis

Using the MIMIC-III dataset, we analyzed patient visit records as reference points and randomly selected 10,000 samples to calculate diagnosis and drug similarities with the Jaccard index. The analysis revealed a Pearson correlation coefficient of 0.53 between diagnosis similarity and drug recommendation similarity, indicating a moderately positive correlation. This finding reinforces the idea that drug recommendation models can align closely with clinical practices by leveraging patterns in patient diagnosis data, as shown in Figure 4 (a). To further illustrate this point, we additionally examined the similarity between patient diagnoses and recommended drugs. The results, summarized in Table 4, clearly show that higher diagnosis similarity corresponds to a greater overlap in drug recommendations.

### F.3  Limited Scene Analysis

In scenarios with limited patient visits or sparse longitudinal history, models relying heavily on memorized patterns from past records tend to degrade in performance. For instance, COGNet's

Table 4: Relationship between diagnosis similarity and drug recommendation overlap.

| Diagnosis Similarity Range | 0.0–0.1 | 0.1–0.2 | 0.2–0.3 | 0.3–0.4 | 0.4–0.5 |
|---|---|---|---|---|---|
| Pair Count | 2641 | 1820 | 1681 | 1400 | 1023 |
| Avg. Jaccard (Drug Rec.) | 0.2835 | 0.2945 | 0.3047 | 0.3155 | 0.3222 |

| Diagnosis Similarity Range | 0.5–0.6 | 0.6–0.7 | 0.7–0.8 | 0.8–0.9 | 0.9–1.0 |
|---|---|---|---|---|---|
| Pair Count | 672 | 411 | 206 | 108 | 38 |
| Avg. Jaccard (Drug Rec.) | 0.3367 | 0.3541 | 0.3873 | 0.4140 | 0.5067 |

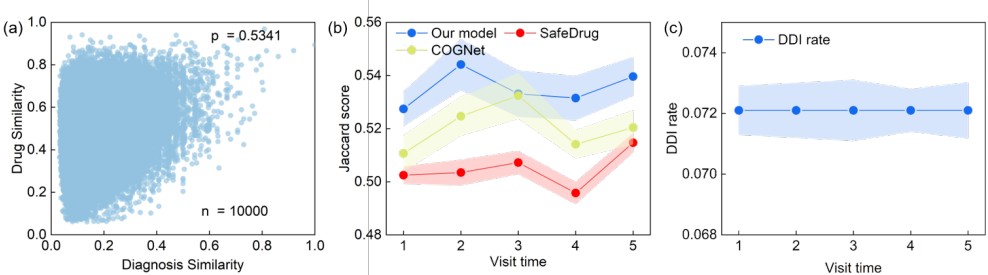

Figure 4: Model performance on a limited number of patient visits scenarios.

"copy-or-predict" mechanism can be effective when sufficient history exists, but it suffers when contextual information is scarce. Under these constraints, our evaluation on the MIMIC-III dataset shows that SubRec maintains consistently low and stable DDI rates (Figure 4 (b), (c)), underscoring its ability to ensure patient safety while providing effective recommendations. Extremely, in a single-drug recommendation setting in Table 5, where each visit results in only one prescribed drug, historical signals are insufficient to support accurate predictions., Compared to COGNet, which exhibits the highest memory consumption and relies on historical repetition, SubRec leverages a structure-aware design to extract condition-specific drug substructures via the GIB module and further refines them through the VQ mechanism. This enables efficient similarity matching and ensures that the learned representations remain both clinically relevant and robust, even with restricted historical input. As a result, SubRec outperforms other baselines, achieving superior accuracy relative to SafeDrug and MoleRec.

## G   Additional sensitivity analyses

As shown in Table 6, increasing the number of vectors in the codebook directly leads to a rise in model parameters, memory consumption, and training time. Given the vast potential space and the diversity of patient-drug combinations, our objective is to make the latent codes compact, meaningful, and represented with as few bits as possible. Simultaneously, it is crucial for the latent codes to convey as much information as possible, ensuring the model remains confident in the latent codes derived from the input. Thus, balancing the trade-off between description length and information content in the latent code becomes essential.

## H   Case study

Here, we present the dimensionality reduction analysis results of the codebook vectors in SubRec, as shown in Figure 6. The training samples are effectively clustered, with the environment embeddings serving as cluster centers. Further analysis of cluster center embeddings from category 5 reveals that they correspond to the diagnoses recorded in Visit 30, alongside their associated prescribed medications. A closely related vector, Visit 37, shows a high degree of overlap in diagnostic records (red color), with a corresponding similarity in prescribed drugs.

Moreover, Visit 44, a test sample vector, is assigned to category 5 based on calculations from Equation (17). Visualization reveals that its diagnoses and drug recommendation results are highly

Table 5: Performance comparison under single-drug recommendation scenario.

| Model | Jaccard |
|---|---|
| SubRec | **0.1620 ± 0.1641** |
| MoleRec | 0.1269 ± 0.1108 |
| COGNet | 0.1037 ± 0.0086 |
| SafeDrug | 0.0755 ± 0.0043 |
| GameNet | 0.0521 ± 0.0032 |

Table 6: Performance with different codebook sizes.

| Size | Time (min) | Memory (MB) | Params |
|---|---|---|---|
| 16 | 392.95 | 1762 | 2.96M |
| 32 | 459.45 | 1762 | 2.97M |
| 2000 | 582.42 | 1770 | 3.48M |
| 5000 | 580.94 | 1782 | 4.26M |
| 20000 | 975.18 | 1864 | 8.14M |

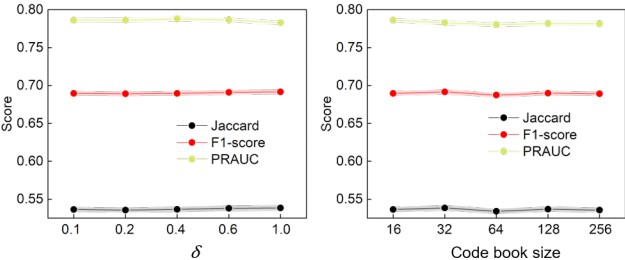

Figure 5: Sensitivity analysis results for different hyperparameters $\delta$ and codebook size $S$.

similar to those of Visit 30 (green color). This indicates that the VQ-based codebook encoding effectively captures and stores representative patient-drug relationships, enabling the model to achieve accurate drug recommendations.

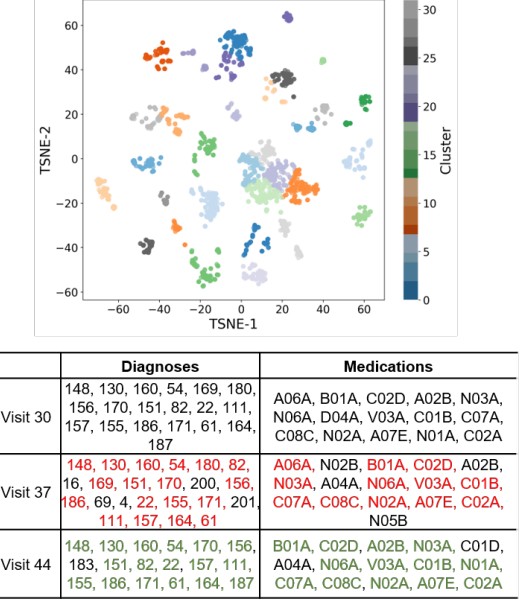

| | Diagnoses | Medications |
|---|---|---|
| Visit 30 | 148, 130, 160, 54, 169, 180, 156, 170, 151, 82, 22, 111, 157, 155, 186, 171, 61, 164, 187 | A06A, B01A, C02D, A02B, N03A, N06A, D04A, V03A, C01B, C07A, C08C, N02A, A07E, N01A, C02A |
| Visit 37 | 148, 130, 160, 54, 180, 82, 16, 169, 151, 170, 200, 156, 186, 69, 4, 22, 155, 171, 201, 111, 157, 164, 61 | A06A, N02B, B01A, C02D, A02B, N03A, A04A, N06A, V03A, C01B, C07A, C08C, N02A, A07E, C02A, N05B |
| Visit 44 | 148, 130, 160, 54, 170, 156, 183, 151, 82, 22, 157, 111, 155, 186, 171, 61, 164, 187 | B01A, C02D, A02B, N03A, C01D, A04A, N06A, V03A, C01B, N01A, C07A, C08C, N02A, A07E, C02A |

Figure 6: TSNE analysis of codebook vectors in SubRec codebook. The table below displays the patient records, with red indicating the overlap of diagnoses and drugs between training sample vectors and cluster center vectors, and green indicating the overlap between test sample vectors and cluster center vectors.

# I Complexity Analysis

While our method introduces some additional computational cost compared to lightweight baselines, the overhead remains manageable and can be effectively mitigated through practical strategies. As shown in Table 7, SubRec consumes less memory than COGNet and is only marginally slower than

Table 7: Comparison of model efficiency in terms of memory, parameters, training, and inference (Shared GPU, NVIDIA Tesla V100 16G).

| Model | Memory (MB) | Para. (M) | Train (h) | Test (10 runs; min) |
|---|---|---|---|---|
| GameNet | 496 | 0.44 | 5.33 | 1.22 |
| SafeDrug | 1716 | 0.37 | 4.06 | 0.45 |
| MoleRec | 1422 | 0.51 | 10.71 | 0.85 |
| COGNet | 3266 | 1.36 | 24.44 | 6.44 |
| SubRec (codebook size = 32) | 1762 | 2.97 | 20.63 | 4.50 |

Table 8: Efficiency improvements of SubRec under different strategies.

| Variant | Train (h) | Test (10 runs; min) |
|---|---|---|
| SubRec (Shared GPU) | 20.63 | 4.50 |
| SubRec (Early stopping) | 8.71 | 4.01 |
| SubRec (Dedicated GPU) | 7.23 | 2.07 |

SafeDrug and MoleRec, while achieving substantially better predictive accuracy. The increased complexity largely stems from modeling the sparsity of longitudinal EHRs and the heterogeneity of drug structures—challenges that prior methods struggle to address. For instance, COGNet exhibits the highest memory consumption due to its reliance on historical information, whereas GameNet adopts a lookup-based mechanism that falls outside the scope of deep learning. In contrast, SubRec introduces a VQ module to cluster heterogeneous patient–drug interactions into a compact codebook, mitigating the instability of variational modeling and enabling efficient similarity matching. To further reduce runtime, we evaluate optimization strategies (Table 8); applying early stopping yields over 2× training speedup with negligible performance loss, and a dedicated GPU environment reduces training time to about 7 hours. Beyond efficiency, SubRec improves interpretability by incorporating an enhanced CIB to extract pharmacologically meaningful substructures conditioned on patient context, while the VQ mechanism enforces a discrete latent structure that preserves only the most relevant features. This design overcomes the limitations of prior rule-based fragmentation approaches (e.g., BRICS in SafeDrug and MoleRec), which disrupt functional group connectivity and overlook the synergistic effects of substructures. As pharmacological studies have shown, many drugs act through shared pharmacophores (e.g., simvastatin and atorvastatin targeting HMG-CoA reductase), and reactive substructures often explain adverse drug interactions. SubRec explicitly captures such motifs, balancing efficiency, accuracy, and interpretability to provide personalized and clinically meaningful recommendations.

## J   Boarder Impacts Statements

The proposed framework, **SubRec**, advances personalized medicine by integrating longitudinal electronic health records (EHRs) and molecular substructure knowledge for precise and safe drug recommendations. Future societal consequences include the potential to democratize access to advanced healthcare technologies by enabling scalable and cost-effective drug recommendation systems across diverse healthcare settings. Additionally, the incorporation of molecular substructure knowledge opens new pathways for drug repurposing and accelerated drug discovery, further benefiting global healthcare. By aligning AI systems with clinical practices and ethical standards, SubRec holds promise to not only improve individual patient outcomes but also transform the broader landscape of medical treatment and drug safety.

## K   Limitation

While SubRec demonstrates strong performance in personalized drug recommendation, it has several limitations. First, the learned discrete codebook reflects the latent space of the training dataset and thus provides strong adaptability within this space. However, for out-of-distribution (OOD) patient cases—whose profiles deviate significantly from the training distribution—the model may require retraining or codebook extension to maintain performance. Second, the framework assumes access to detailed molecular graph information for all candidate drugs, which may not always be available in

practical clinical settings. Future work could explore dynamic codebook updates and alternative drug representations to improve generalizability and scalability.

