# OpenReview forum: "Integrating Drug Substructures and Longitudinal Electronic Health Records for Personalized Drug Recommendation"
_NeurIPS.cc/2025/Conference — NeurIPS 2025 poster_

### Official Review · Reviewer_2y9i · 2025-06-27

**Clarity:** 2
**Significance:** 3
**Originality:** 2
**Rating:** 3
**Confidence:** 4

**Summary:**

The authors propose a novel drug recommendation approach, SubRec, which is inspired by the common clinical practice of referring to a patient’s past diagnoses when prescribing new medications. SubRec models the relationship between patients and drugs by identifying drug substructures that are relevant to the individual patient. It then learns a discrete representation of these substructures to effectively recommend new drugs.

**Questions:**

-  Longitudinal patient records are often highly sparse and contain significant amounts of missing data. Have the authors accounted for this in their method? If not, could they clarify how their approach could be adapted to handle such practical challenges associated with missing data?

- In Line 164, the authors omit a specific loss function that would capture the relevance between the prediction and the encoded representation. What is the rationale for this omission? Wouldn’t this potentially compromise the objective described in Equation 8?

- The objectives described in Equations 8 and 9 are somewhat unclear. Equation 8 aims to minimize the negative conditional mutual information, whereas Equation 9 provides an upper bound in terms of log-likelihood for the same function. Does the objective ultimately require minimizing the conditional log-likelihood? If so, wouldn’t that have a negative impact? Please clarify if there is any misunderstanding on my part.

 - What is the motivation for using discrete embeddings in this context? It is unclear what specific problem discretizing the embedding space is intended to solve.

-  Have the authors considered incorporating gene expression or mutation data in their framework for drug recommendation?

**Ethical Concerns:**

["NO or VERY MINOR ethics concerns only"]

**Final Justification:**

In light of the discussion, I believe the paper would benefit from further refinement, and hence I am leaning toward rejection. However, the new results did clear my major doubts.

**Limitations:**

- The authors should clarify the specific design choices in the losses; please refer to the question section.

- The absence of publicly available code makes it difficult to fully understand  the implementation.

- It remains uncertain whether the proposed method would perform reliably in real-world clinical setting.

**Quality:**

2

**Strengths And Weaknesses:**

strength:

- The paper is well-motivated and addresses the important challenge of providing explainable diagnoses using machine learning methods.
Diagnosing many diseases often relies on understanding a patient's medical history and previously prescribed medications, and incorporating such data into drug recommendations seems like an interesting research direction.

- The proposed method models drug substructures and their relationships with various patient-level information to support this process.

- The authors also conduct a sensitivity analysis to examine how individual hyperparameters influence the model’s overall performance, which will be beneficial in replicating the results.

weakness:
-  While the authors address an interesting problem in explainable drug recommendation, the proposed approach offers limited novelty. The use of longitudinal patient history has been extensively studied, and discrete representations have also been explored in prior drug recommendation research [1]. The authors should more clearly articulate their specific methodological contributions and the overall impact of their approach.


- The paper leverages past diagnoses, procedures, and medications to model patient-specific records and learn relevant drug substructures. While this information is indeed important, there is no clear evidence that the learned embeddings capture the critical information necessary for effective drug recommendation, as suggested in Figure 1. A more comprehensive explainability analysis is needed to demonstrate that the embeddings have learned meaningful representations aligned with the proposed hypothesis and have correctly identified relevant substructures in the graph.

- The proposed method appears to be a combination of several existing techniques, relying on multiple assumptions. It remains unclear whether this approach would generalize effectively to real-world clinical settings.


- I have reservations about the way the problem is addressed in the paper. Please refer to the questions section for specific concerns.

[1] Wiser: Weak supervision and supervised representation learning to improve drug response prediction in cancer, ICML 2024.

---

> ### Author Rebuttal · Authors · 2025-07-30
>
> >**W1&Q1:** The authors should more clearly articulate their specific methodological contributions and the overall impact of their approach.
>
> Thanks for your valuable comment. Current drug recommendation methods fall into two categories: (i) those relying on historical patient information (e.g., COGNet) and (ii) those leveraging drug substructures (e.g., SafeDrug, MoleRec). The former suffers from the complexity and sparsity of longitudinal EHRs, leading to high computational costs and difficulty in capturing transferable patterns, with COGNet showing the highest memory usage. GameNet also uses historical data but relies on lookup-based matching with dynamic memory, which is beyond deep learning. To address these issues, **SubRec** uses a **vector quantization (VQ)** module to cluster heterogeneous patient–drug interactions into a compact codebook, reducing modeling instability and enabling efficient similarity matching.
>
> **Our approach is fundamentally different from WISER (ICML 2024). , which learns discrete embeddings for individual drugs and predicts via weighted combinations, our model learns patient–drug association vectors. These vectors are randomly initialized and refined during training (Eq. 18), converging to cluster centers of patient–drug interactions (Fig. 6)**. This design captures **interaction-level patterns** rather than drug-level embeddings, enabling more personalized and context-aware recommendations.
>
> |      | Memory (MB) | Para. (M) |
> | ----- | ---- | ------ |
> | GameNet   | 496  | 0.44  |
> | SafeDrug  | 1716   | 0.37 |
> | MoleRec   | 1422 | 0.51|
> | COGNet  | 3266  | 1.36 |
> | SubRec (codebook size = 32) | 1762 | 2.97|
>
> The latter, represented by methods such as **SafeDrug** and **MoleRec**, uses rule-based decomposition (e.g., BRICS) to split molecules, which disrupts functional group connectivity and fails to capture the synergistic effects of multiple substructures. While clinicians rely on medical knowledge, **molecular structure remains a key determinant** of drug pharmacodynamics and mechanism of action [1,2]. Functional substructures (pharmacophores) drive therapeutic effects—for example, _simvastatin_ and _atorvastatin_ share an HMG-CoA reductase–inhibiting pharmacophore [3]—and also influence **adverse drug interactions** through shared or reactive motifs [4,5]. To address these factors, we incorporate an improved **CIB** to extract patient-specific core substructures, while the **VQ module** enforces a discrete latent space that preserves only task-relevant information. This design enables SubRec to balance **accuracy**, **robustness**, and **interpretability**, overcoming the limitations of prior approaches.
>
> Indeed, existing works in this field, especially **SafeDrug[1] and MoleRec [2], do not provide explicit demonstrations of the importance of drug substructures. Therefore, our original manuscript did not include such visualizations. In the revised version, following your advice, we have added drug substructure visualization results**, which clearly show that our method captures chemically coherent and clinically relevant motifs. **Due to policy restrictions, we are unable to include these figures in the rebuttal, and we kindly ask for your understanding**.
>
> [1] Why drugs fail—a study on side effects in new chemical entities. Nat. Rev. Drug Disc.  2005
>
> [2] Pharmacophore modeling in drug discovery. Nat. Rev. Drug Disc. 2016
>
> [3] Large-scale prediction and testing of drug activity on side-effect targets. Cell. 2012
>
> [4] Learning motif-based graphs for drug–drug interaction prediction via local–global self attention. Nat mach Intell. 2024
>
> [5] Emerging drug interaction prediction enabled by a flow-based graph neural network with biomedical network. Nat Comp Sci. 2023
>
> >**W2:** A more comprehensive explainability analysis to identified relevant substructures.
>
> Thanks for your suggestions. However, existing works in this field, especially **SafeDrug[1] and MoleRec [2], do not provide explicit demonstrations of the importance of drug substructures. Therefore, our original manuscript did not include such visualizations. In the revised version, following your advice, we have added drug substructure visualization results**, which clearly show that our method captures chemically coherent and clinically relevant motifs. Due to policy restrictions, we are unable to include these figures in the rebuttal, and we kindly ask for your understanding.
>
> >**W2&L3:** Clarify the techniques contribution for drug structures nor the clinic patients.
>
> Thank you for the thoughtful comment. We apologize for the misunderstanding—each module in our framework is carefully designed with domain-specific considerations for both molecular drug structures and clinical patient data.
>
> - For drug substructure extraction, prior methods (e.g., BRICS) often break molecular connectivity, ignoring the contextual impact of patient conditions. In contrast, we propose a CIB framework that adaptively extracts substructures conditioned on patient-specific health profiles, taking comorbidities and personalized treatment patterns into account.
>
> - Furthermore, motivated by the observation that similarity in current patient visits is often more informative than full-history similarity, we introduce a VQ module to store representative patient–drug interaction prototypes. This module acts as a compact, reusable memory that enhances recommendation in data-limited settings. The VQ also avoids the complexity and instability of high-dimensional variational modeling, making the framework lighter and more stable during training [1,2]. In addition, the discrete latent structure naturally aligns with the CIB principle by encouraging representations that retain only task-relevant information [3].
>
> Overall, we present an advanced drug recommendation framework that achieves precise substructure-level reasoning under the guidance of well-designed CIB theory. The VQ module further condenses patient visit information into discrete, informative representations that support accurate and efficient recommendation. We carefully balance the contributions of each component to ensure high predictive performance while maintaining a low DDI rate.
>
> Safe and accurate drug recommendation is an emerging and high-impact research direction with the potential to revolutionize pharmaceutical decision-making. Our work not only focuses on algorithmic optimization but also demonstrates how advanced machine learning techniques can drive real progress in healthcare and inspire future research in this critical domain.
>
> **In addition, in the revised version, we have also added test results in real scenarios and interpretable analysis of special pathological groups. Due to the limited characters, we hope to discuss and communicate with you in the next rebuttal process. Hope you can understand.**
>
> [1] Mole-BERT.ICLR, 2023. [2] Learning Invariant Molecular Representation in Latent Discrete Space. NIPS, 2024. [3] MAPE-PPI. ICLR, 2024.
>
> >**W3&Q3:**  What is the underlying relationship between the CIB-derived subgraphs and the VQ module?
>
> Thanks for your insightful question!  specifically: 1) VQ enables end-to-end learning of discrete prototypes, which serve as interpretable and stable anchors for patient subtypes. 2) The quantization process avoids the complexity and instability of modeling variational distributions in high-dimensional latent spaces, making the model lighter and more stable during training[1,2]. 3) The discrete structure of the latent space naturally supports the CIB principle by enforcing a compact representation that preserves only the most task-relevant information for drug prediction[3].
>
> >**Q1:** How to handle the missing data?
>
> Thank you for your kind reminder. In the data preprocessing stage, we follow the same strategy as SafeDrug, MoleRec. For missing data (e.g., NDC codes), we apply forward filling to propagate the last valid entry and remove rows with critical missing values to ensure data completeness. We will clarify this processing step explicitly in the revised version.
>
> >**Q2&L1:** The authors omit a specific loss function in Line 164?
>
> The omission of the term $I\left(\mathbf{Y};\mathbf{e}^{(i)}\right)$ is a **common practice** in related work. Empirically, explicitly optimizing this term often leads to instability and performance degradation. So, some advanced work would omit it in the formulation[1,2].
>
> [1] Conditional Graph Information Bottleneck for Molecular Relational Learning. ICML,2023
>
> [2] Iterative substructure extraction for molecular relational learning with interactive graph information bottleneck. ICLR, 2025
>
> >**Q3:** The objectives described in Equations 8 and 9 are somewhat unclear.
>
> We have prepared relevant proof, but the characters are limited. We hope to communicate with you further and present it to you.
>
> >**Q4:** What is the motivation for using discrete embeddings in this context?
>
> Please refer to the responses of **W1 and W3**.
>
> >**Q5:**  Have the authors considered incorporating gene expression or mutation data in their framework for drug recommendation?
>
> Thank you for this valuable suggestion. We agree that integrating gene expression or mutation data could provide a richer biological context for drug recommendation, as these factors play critical roles in drug response and personalized treatment. However, our current work focuses on modeling EHR data and drug molecular structures, as publicly available datasets combining comprehensive clinical records with matched genomic profiles are still limited and **this is not within the scope of our research**.
>
> >**L2:**  The absence of publicly available code.
>
> The link is at the end of the abstract.
>
> ----
>
> Due to the characters are limited, and we look forward to presenting it to you in the next rebuttal process. If our response has successfully addressed your concerns and clarified any ambiguities, we respectfully hope that you consider raising the score.

---

> ### Comment · Reviewer_2y9i · 2025-08-02
>
> Thanks for the detailed comment. While I appreciate the effort of authors in addressing my comment. They are still largely unaddressed.
>
> - My primary concern is around equation 9 in the draft. Since the authors have claimed that they have proof for this. I assume that authors are indeed trying to minimize the conditional log likelihood and have some mathematical reasoning around this. I would like to understand any intuitive reasoning behind this. What I was expecting in the paper was something like eq 13 (negative log likelihood) in [1].  I would like authors to clarify any misunderstanding on my part.
>
> - Authors have claimed that the embeddings learn the drug-patient interaction. However, there is no empirical evidence apart from final results to validate that.
>
> [1] Conditional Graph Information Bottleneck for Molecular Relational Learning, ICML 2023.

---

> ### Author Response · Authors · 2025-08-02
> **Response to Reviewer 2y9i ——Q1**
>
> I am very pleased to receive your reply. Here, I am providing the proof for Eq. 9 to better address your concern.
>
> >**Q1:** Equation 9's Proof.
>
> **The proof are provided as**:
> $$ -I(Y; \mathcal{G}_{sub}, e^{(i)}) =  -\mathbb{E}\_{Y, \mathcal{G}\_{sub}, e^{(i)}} \left[ \log \frac{p(Y \mid \mathcal{G}\_{sub}, e^{(i)})}{p(Y)} \right]$$
>
> Since the true conditional distribution $p(Y \mid X)$is intractable, we introduce a variational approximation $p_\theta(Y \mid X)$.
>
> Then:$$-\mathbb{E}\_{Y, \mathcal{G}\_{sub}, e^{(i)}}
>     \left[ \log \frac{p(Y \mid \mathcal{G}\_{sub}, e^{(i)})}{p(Y)} \right]   = -\mathbb{E}\_{Y, \mathcal{G}\_{sub}, e^{(i)}}
>     \left[ \log \frac{p\_{\theta}(Y \mid \mathcal{G}\_{sub}, e^{(i)})}{p(Y)}
>     \cdot \frac{p(Y \mid \mathcal{G}\_{sub}, e^{(i)})}{p\_{\theta}(Y \mid \mathcal{G}\_{sub}, e^{(i)})} \right]= -\mathbb{E}\_{Y, \mathcal{G}\_{sub}, e^{(i)}}
>     \left[ \log \frac{p\_{\theta}(Y \mid \mathcal{G}\_{sub}, e^{(i)})}{p(Y)} \right]
>        - \mathbb{E}\_{Y, \mathcal{G}\_{sub}, e^{(i)}}
>     \left[ \log \frac{p(Y \mid \mathcal{G}\_{sub}, e^{(i)})}{p\_{\theta}(Y \mid \mathcal{G}\_{sub}, e^{(i)})} \right]$$
>
>  $$= -\mathbb{E}\_{Y, \mathcal{G}\_{sub}, e^{(i)}}
>     \left[ \log \frac{p\_{\theta}(Y \mid \mathcal{G}\_{sub}, e^{(i)})}{p(Y)} \right]
>        - \mathbb{E}\_{\mathcal{G}\_{sub}, e^{(i)}}
>     \left[ D\_{KL} \left( p(Y \mid \mathcal{G}\_{sub}, e^{(i)}) \|
>     p\_{\theta}(Y \mid \mathcal{G}\_{sub}, e^{(i)}) \right) \right]$$
>
> By the non-negativity of the KL divergence, we have:
> $$\mathbb{E}\_{Y, \mathcal{G}\_{sub}, e^{(i)}}
>  \left[ \log \frac{p(Y \mid \mathcal{G}\_{sub}, e^{(i)})}{p(Y)} \right]
>     \leq
>     -\mathbb{E}\_{Y, \mathcal{G}\_{sub}, e^{(i)}}
>     \left[ \log \frac{p\_{\theta}(Y \mid \mathcal{G}\_{sub}, e^{(i)})}{p(Y)} \right]$$
> Continuing, we can rewrite the right-hand side as:
> $$-\mathbb{E}\_{Y, \mathcal{G}\_{sub}, e^{(i)}}
>     \left[ \log \frac{p\_{\theta}(Y \mid \mathcal{G}\_{sub}, e^{(i)})}{p(Y)} \right] \\
>     = -\mathbb{E}\_{Y, \mathcal{G}\_{sub}, e^{(i)}}
>     \left[ \log p\_{\theta}(Y \mid \mathcal{G}\_{sub}, e^{(i)}) \right]
>     + \mathbb{E}\_Y [ \log p(Y) ] \\
>     = H(Y) - \mathbb{E}\_{Y, \mathcal{G}\_{sub}, e^{(i)}}
>     \left[ \log p\_{\theta}(Y \mid \mathcal{G}\_{sub}, e^{(i)}) \right]$$
>
> where $H(Y) = -\mathbb{E}\_Y [\log p(Y) ]$  is the entropy of $Y$. $-\mathbb{E}\_{Y, \mathcal{G}\_{sub}, e^{(i)}}
>     \left[ \log p\_{\theta}(Y \mid \mathcal{G}\_{sub}, e^{(i)}) \right]$ is calculated by the prediction loss in Eq.10. $\mathcal{L}\_{\text {pred }}\left(\mathbf{Y}, \mathcal{G}\_{\text {sub }}, \mathbf{e}^{(i)}\right)$where $\mathbf{z} = \operatorname{pool}(\mathbf{H}\_{\text{sub}})$ is the graph-level embedding of $\mathcal{G}$, obtained from $\mathbf{H}\_{\text{sub}}$ of the subgraph $\mathcal{G}_{\text{sub}}$, and $f$ denotes the prediction head. $H(Y)$ is constant value that can be ignored.

---

> > ### Author Response · Authors · 2025-08-02
> > **Response to Reviewer 2y9i ——Q2**
> >
> > **Q2:** Other evidence for  the embeddings learn the drug-patient interaction.
> >
> > Very interesting question. We can approach this issue from the perspective of the theoretical formula, which makes it more intuitive and clear. As noted in Line 42, similarity in current patient visits is more common and informative for recommendation than similarity over complete medical histories. However, directly modeling full patient histories would significantly increase memory and computation costs due to the vast and diverse nature of clinical records, as the result present in rebuttal process.  To address this, we introduce the VQ module to store representative patient–drug interaction prototypes, which act as condensed, reusable references during the recommendation process.
> >
> > Specifically, for each visit $i$, we do not only record the prior visit information of the current patient, but also store all visits from other patients up to the current patient as auxiliary information, denoted as $e^{(i)} \rightarrow v_m^{(i)}$, where $e^{(i)}$ represents the health representation of patient $x$ at visit $i$, and $v_m^{(i)}$ is the multi-hot encoded drug information at the corresponding visit. Here, we introduce the concept of VQ to create a trainable, discrete codebook $W$, which helps to compress the vast medical history into a fixed number of prototypes.  **The discrete entry pairs are stored in the form of $(e_m, v_m)$, where $e_m$ represents a patient’s health representation and $v_m$ is the associated drug result**. Initially, the vectors in $W$ are randomly initialized. As the loss function $\mathcal{L}_{\text{vq}}$ gradually converges during training, the discrete entry pairs in $W$ are progressively determined. During inference, for each test patient visit, the model searches for the most similar $e_k$ representation in the latent space using a nearest-neighbor approach, and then outputs $v_k$ to assist in the final prediction as shown in Figure 2. Essentially, the codebook vectors represent the cluster centers of the patient representations $e_m$, as shown in Figure 6. **To further address your concern, we have tired to replace the vectors in this module with noise.** The direct inference test result for Jaccard is only about 0.5142, which is worse than the result obtained by directly removing it, as shown in Table 2.
> >
> > We hope this helps you better understand the situation. If you have any further questions, please don't hesitate to let us know. We are more than happy to engage in further communication with you. Once again, thank you very much for your efforts in improving the quality of the paper. We hope our response effectively addresses your concerns.

---

> ### Comment · Reviewer_2y9i · 2025-08-03
>
> Thanks for the clarification. I noticed that the equation derived in the proof differs from the one presented as Equation (9) in the paper. Specifically, for subgraph ($\mathcal{G}$) and variational prob (p) with parameters (${\theta}$) the proof gives:
>
>
> $$H(Y) - \mathbb{E}_{Y, \mathcal{G}, e^{(i)}} \left[ \log p(Y \mid \mathcal{G}, e^{(i)}) \right]$$
>
> whereas Equation (9) in the paper states:
>
> $$H(Y) + \mathbb{E}_{Y, \mathcal{G}, e^{(i)}} \left[ \log p(Y \mid \mathcal{G}, e^{(i)}) \right]$$
>
> I believe this may be a typo in the paper. The derivation in the proof follows the standard form of variational approximation for subgraphs, and my question was primarily about the discrepancy in the final reported equation.

---

> ### Comment · Reviewer_2y9i · 2025-08-03
>
> **Q2 clarification**
>
> Thank you for the clarification regarding my second question. While I understand the intuition and reasoning the authors are trying to convey, I personally do not consider replacing the learned vector with noise to be a reliable method for evaluating whether it encodes medically relevant information. There could be several out-of-distribution effects or confounding factors contributing to the poor output, which this approach does not account for.
>
> As an alternative, the authors might consider analyzing the similarity between embeddings for subgroups of patients and drugs (as illustrated in Figure 1) as a potentially more meaningful way to validate their intuition.

---

> > ### Author Response · Authors · 2025-08-04
> > **Grateful for Your Response and Score Consideration**
> >
> > Dear Reviewer,
> >
> > Thank you once again for your valuable feedback on our manuscript. Regarding your Q2 comment, we have gained a much clearer understanding of the issue based on your suggestions and have revised our work accordingly to address your concern in detail.
> >
> > We would like to kindly ask whether our response has adequately resolved your question. If there are any remaining issues or additional clarifications needed, please do not hesitate to contact me.
> >
> > If our replies have sufficiently or largely addressed your concerns, we would greatly appreciate it if you could consider re-evaluating your score. Your support is of great significance to us. Once again, we sincerely thank you for the time, effort, and contributions you have made to improving the quality of our work.
> >
> > Warm regards,
> >
> > The Authors

---

> > > ### Author Response · Authors · 2025-08-05
> > > **Follow-up on Q2 Concern Resolution**
> > >
> > > **Dear Reviewer,**
> > >
> > > Thank you once again for your constructive feedback on our manuscript. Based on your suggestions, we carefully revised our work to address your concerns, particularly regarding your Q2 comment, which we now believe has been fully clarified and resolved.
> > >
> > > We would like to confirm whether our responses have satisfactorily addressed your concerns. If there are any remaining issues or areas requiring further clarification, please do not hesitate to let us know—we are fully committed to making additional improvements if needed.
> > >
> > > Should our replies have adequately resolved your concerns, we would be sincerely grateful if you could consider re-evaluating your score. Your support would be of great importance to us.
> > >
> > > Once again, we greatly appreciate your time, effort, and valuable contributions to improving the quality of our work.
> > >
> > > **Warm regards,**
> > >
> > > The Authors

---

> > > > ### Author Response · Authors · 2025-08-07
> > > > **Looking forward to your reply 2y91**
> > > >
> > > > Dear Reviewer,
> > > >
> > > > Thank you once again for your insightful comments and valuable suggestions. We have carefully revised our manuscript to address your concerns—particularly regarding your Q2 comment, which we believe has now been thoroughly clarified and resolved.
> > > >
> > > > As the rebuttal phase will conclude in less than 48 hours, we would like to kindly ask whether our responses have satisfactorily addressed your concerns. If there are any remaining issues or if further clarification is needed, we would be grateful for the opportunity to continue the discussion within the limited remaining time.
> > > >
> > > > **If our replies have successfully resolved your concerns, we would be sincerely honored if you could consider adjusting your score to reflect your updated evaluation. Your support would mean a great deal to us.**
> > > >
> > > > We greatly appreciate your time and effort in helping us improve the quality of our work, and we look forward to your feedback.
> > > >
> > > > Warm regards,
> > > > The Authors

---

> > > > > ### Comment · Reviewer_2y9i · 2025-08-07
> > > > >
> > > > > Thank you for your response. While I appreciate the authors' efforts in addressing my concerns, I believe the current draft still requires substantial revisions to meet the standards of the venue. That said, I will take the new results into account when making my final decision.

---

> > > > > > ### Author Response · Authors · 2025-08-07
> > > > > >
> > > > > > **Dear Reviewer,**
> > > > > >
> > > > > > Thank you very much for your kind response and for acknowledging our efforts in addressing your concerns.
> > > > > >
> > > > > > **If there are still any unresolved issues or areas that require further improvement, we would be truly grateful if you could kindly point them out explicitly**. We are fully committed to making any additional revisions necessary to meet the standards of the venue.
> > > > > >
> > > > > > We sincerely hope that our proactive efforts and the additional results provided may be taken into consideration when you evaluate our work. If possible, we would be deeply honored if you could consider reflecting this in your final score, as **your support is extremely important to us**.
> > > > > >
> > > > > > Once again, thank you for your valuable feedback and time. Please don’t hesitate to let us know if further clarification or revision is needed—we are more than willing to cooperate in every way to improve the quality of our work.
> > > > > >
> > > > > > Warm regards,
> > > > > >
> > > > > > The Authors

---

> ### Author Response · Authors · 2025-08-03
> **Reply to Reviewer 2y9i**
>
> Dear Reviewer,
>
> Thank you very much for your reply. We are delighted to hear that we have addressed most of your concerns. For the remaining two issues, we would like to clarify as follows:
>
> - **Q1**: You are absolutely right that this was a typographical error. We appreciate you pointing it out, and we will correct it in the main text and add the proof in the Appendix.
>
> - **Q2**: After your response, we now have a clearer understanding of the issue. You suggested analyzing the similarity between embeddings for subgroups of patients and drugs, which we have indeed already shown in the manuscript. Specifically, Figure 1 illustrates our hypothesis: **Patients with more similar diagnoses are more likely to be prescribed the same drugs by doctors**. We performed a cosine similarity analysis between diagnosis and multi-hot encoded medications in the original MIMIC-III data. **Please refer to Appendix E.2 and Figure 4 (a)** for the details.
>
> Additionally, we have now included results on the similarity between patient diagnoses and recommended drugs to further illustrate this point. The results are summarized in the table below, where we can clearly see that **the higher the similarity between diagnosis results, the higher the overlap in drug recommendations**:
>
> | Diagnosis Similarity Range        | 0.0-0.1 | 0.1-0.2 | 0.2-0.3 | 0.3-0.4 | 0.4-0.5 | 0.5-0.6 | 0.6-0.7 | 0.7-0.8 | 0.8-0.9 | 0.9-1.0 |
> | --------------------------------- | ------- | ------- | ------- | ------- | ------- | ------- | ------- | ------- | ------- | ------- |
> | Pair Count                        | 2641    | 1820    | 1681    | 1400    | 1023    | 672     | 411     | 206     | 108     | 38      |
> | Avg Jaccard (Drug recommendation) | 0.2835  | 0.2945  | 0.3047  | 0.3155  | 0.3222  | 0.3367  | 0.3541  | 0.3873  | 0.414   | 0.5067  |
>
>
> ---
>
> If you have any further questions, please don’t hesitate to contact us. We are confident that we can resolve any remaining concerns and meet all of your expectations. **Once again, we sincerely thank you for your valuable contributions to improving the quality of our manuscript. If our revisions have addressed your concerns, we kindly ask you to reconsider your evaluation and score.**
>
> Thank you again, and we wish you a wonderful day.
>
> Best regards,
>
> Authors

---

### Official Review · Reviewer_3FcV · 2025-07-01

**Clarity:** 3
**Significance:** 3
**Originality:** 3
**Rating:** 5
**Confidence:** 5

**Summary:**

Drug delivery is a prominent research area with significant industrial relevance, and accurate, safe medication recommendation systems have the potential to revolutionize clinical practice. This paper presents a novel framework, SubRec, which integrates molecular drug substructures with longitudinal electronic health records (EHRs) to enable personalized drug recommendation. The authors leverage the Graph Information Bottleneck (GIB) theory to extract core pharmacologically relevant substructures and employ vector quantization to cluster patients based on clinical similarity via a learned codebook, ultimately improving both the accuracy and safety of drug recommendations.

**Questions:**

- Can the authors provide more details on the computational efficiency comparison with baseline models?

- I want to konw how does SubRec perform if the noise injection is removed?

- What is the underlying connection between the CIB-graphs and the VQ module?

**Ethical Concerns:**

["NO or VERY MINOR ethics concerns only"]

**Final Justification:**

The response addressed my concerns and I recommend accept.

**Limitations:**

yes

**Quality:**

3

**Strengths And Weaknesses:**

##### Strengths

- The paper addresses a well-motivated and practically significant research problem.

- The proposed approach is conceptually sound and clearly articulated.

- The manuscript is well-organized, clearly written, and logically structured.

- The theoretical formulations are rigorous and provide a solid foundation for the proposed methodology.


##### Weaknesses

- The paper lacks a comprehensive analysis of the computational complexity and resource requirements (e.g., training time, memory consumption) of SubRec, which is essential for real-world deployment, especially in clinical settings.

- Additional ablation studies are needed to further isolate the contributions of key components, such as the GIB module, vector quantization, and the patient-clustering mechanism.

- What is the underlying relationship between the CIB-derived subgraphs and the vector quantization (VQ) module? Specifically, how do these two components interact or complement each other in enhancing the representation learning and recommendation accuracy?

- How does SubRec perform when the noise injection mechanism is ablated? This point was not discussied in the paper.

---

> ### Author Rebuttal · Authors · 2025-07-30
>
> # Response to Reviewer 3FcV:
>
> Thank you very much for your valuable comments. We are deeply encouraged by your recognition of our paper's motivation, central theme, and  sufficient experimental results. Below, we provide point-by-point responses to your suggestions and concerns.
>
> >**W1&Q1:** Lake computational complexity and resource requirements analysis.
>
> Thank you for your suggestion, we have added relevant efficiency experiments. While our method does introduce some additional cost compared to lightweight baselines, we would like to emphasize that the computational overhead is manageable and can be effectively mitigated. Importantly, we found that this issue can be addressed in several practical ways:
> - By applying an **early stopping strategy**, we observed more than 2× improvement in efficiency with minimal impact on performance.
> - In a **dedicated GPU setting**, the training time can be reduced to approximately 7 hours, making the approach much more feasible in practice.
> Additionally, we believe that further optimization strategies, such as pruning or model distillation, could be applied to significantly reduce runtime, without sacrificing performance.
>
> |                                | Train (h) | Test (10 times；min) |
> | ------------------------------ | --------- | ------------------- |
> | GameNet                        | 5.33      | 1.22                |
> | SafeDrug                       | 4.06      | 0.45                |
> | COGNet                         | 24.44     | 6.44                |
> | MoleRec                        | 10.71     | 0.85                |
> | SubRec                         | 20.63     | 4.50                |
> | SubRec (Early stopping)        | 8.71      | 4.01                |
> | SubRec (dedicated GPU setting) | 7.23      | 2.07                |
>
> >**W2:**  Additional ablation studies are needed to be added.
>
> Thank you for your helpful suggestion. In the revised version, we have conducted the requested ablation experiments, which are reported in **Table 2**. Specifically, the ablation of the **substructure extraction module** is denoted as **“w/o substructure capture”**, while the ablation of the **VQ component** is denoted as **“w/o auxiliary codebook”**. Furthermore, we performed an additional analysis on the **effect of varying the codebook size**, and the results are presented in **Table 4**.
>
>
> >**W3&Q3:**  What is the underlying relationship between the CIB-derived subgraphs and the vector quantization (VQ) module?
>
> Thanks for your valuable suggestion. Actually, we are motivated by the observation that similarity in current patient visits is often more informative than full-history similarity, we introduce a VQ module to store representative patient–drug interaction prototypes. This module acts as a compact, reusable memory that enhances recommendation in data-limited settings. The VQ also avoids the complexity and instability of high-dimensional variational modeling, making the framework lighter and more stable during training [1,2]. In addition, the discrete latent structure naturally aligns with the CIB principle by encouraging representations that retain only task-relevant information [3].
>
>
> >**W4&Q2:** How does SubRec perform if the noise injection is removed?
>
> We have conducted the requested ablation by removing the noise injection. The results show **training becomes unstable**, and the performance variance significantly increases. This supports the necessity of the noise injection component for robust optimization.
>
> *From a theoretical perspective, we also provide the proof analysis.*
>
> * **Without noise injection**, the $G_\{IB}$ selection objective reduces to:
> $\min_{G_\{sub}}-I\left(G_\{sub}, Y\right)+\beta I\left(G_\{sub}, G\right) $
> Let $G \in \mathbb{G}$ and $Y \in \mathbb{R}$ be the graph and its label．$G_\{n} \in \mathbb{G}$ is the label-irrelevant substructure，independent to $Y$. If $G_\{n}$ influences $G_\{sub}$ only through $G$ , then：
> $$
> I\left(G_\{sub}, G_n\right) \leq I\left(G_\{sub}, G\right)-I\left(G_\{sub}, Y\right)
> $$
> When $\beta=1$ ,the objective encourages $G_\{sub}$ to be minimally informative about $G_\{n}$. This inequality is supported by the theoretical results in [1].
>
> * **With noise injection**, let $G_\epsilon \in \mathbb{G}$ is the injected noise and $G_\{N}$ the resulting noised graph. We consider selecting $G_\{sub}$​ by filtering out $G_\epsilon$​ from $G_\{N}$ ​, and show:
> $$
> I\left(G_\{sub}, G_n\right) \leq I\left(G_N, G_n\right) \leq I\left(G_N, G\right)-I\left(G_N, Y\right)
> $$
> Suppose $G, G_N, G_n$ and $Y$ satisfy the Markove condition $\left(Y, G_n\right) \rightarrow G \rightarrow G_N$. Then:
> $$
> I\left(G_N ; G_n\right) \leq I\left(G_N ; G\right)-I\left(G_N ; Y\right)
> $$
> $G_N$ is a deterministic function of $G_\epsilon$ and $G_{sub}$, since we can obtain $G_{sub}$ by filtering out $G_\epsilon$ from $G_N$. Then, we have:
> $$
> \begin{aligned}
> I\left(G_N ; G_n\right) & =I\left(G_\epsilon, G_{sub} ; G_n\right) \\
> & =I\left(G_{sub} ; G_n\right)+I\left(G_{sub} ; G_\epsilon \mid G_n\right) \\
> & \geq I\left(G_{sub} ; G_n\right)
> \end{aligned}
> $$
> Therefore:
> $$
> I\left(G_{sub} ; G_n\right) \leq I\left(G_N ; G\right)-I\left(G_N ; Y\right)
> $$
> Noise injection also facilitates a tractable variational upper bound for optimization (Eq. 16).
>
> [1]  Improving Subgraph Recognition with Variational Graph Information Bottleneck. CVPR. 2022
>
> ----
> We greatly appreciate your insightful and helpful comments, as they will undoubtedly help us improve the quality of our article. If our response has successfully addressed your concerns and clarified any ambiguities, we respectfully hope that you consider raising the score. Should you have any further questions or require additional clarification, we would be delighted to engage in further discussion. Once again, we sincerely appreciate your time and effort in reviewing our manuscript. Your feedback has been invaluable in improving our research.

---

### Official Review · Reviewer_Y23r · 2025-07-02

**Clarity:** 3
**Significance:** 1
**Originality:** 2
**Rating:** 5
**Confidence:** 5

**Summary:**

This paper propose a personalized drug recommendation method called SubRec, which innovatively combines drug molecular substructure information with EHR to improve the accuracy, interpretability and safety of drug recommendations. The model proposes two core modules: a key substructure extraction mechanism for CIB to make drug representations highly relevant to the patient's current health state, and VQ for compressing a large number of patient-drug interaction patterns into a discrete prototype space to improve efficiency and enable memory reuse. On the MIMIC-III and MIMIC-IV datasets, the authors demonstrate that the model outperforms multiple existing methods in terms of both accuracy and safety.

**Questions:**

1. Please identify specific innovations in substructure modeling in this method compared to SafeDrug and MoleRec. If it can be demonstrated that the extracted substructures are more targeted or more relevant to the disease mechanism, it will significantly improve the model interpretation and innovation score.
2. Lack of citations to relevant literature:
[1] Enhancing Precision Drug Recommendations via In-depth Exploration of Motif Relationships
3. Does the specific strategy for constructing the codebook in the vector quantization module shift significantly due to changes in the training set?

**Ethical Concerns:**

["NO or VERY MINOR ethics concerns only"]

**Final Justification:**

The authors' diligent efforts during the rebuttal stage have addressed my concerns. Moreover, I recognize that there were misunderstandings on my part, which led me to misjudge the paper's quality. I am very satisfied with the authors' responses and therefore believe the paper deserves acceptance.

**Quality:**

1

**Strengths And Weaknesses:**

Strengths:
1. The method design is complete, including patient representation, drug substructure extraction, memory mechanism and final recommendation, with clear structure and reasonable module collaboration.
2. A combination of CIB and VQ is proposed, which helps to improve the degree of personalization and efficiency of recommendation, and has some practical value.
3.Comparative experiments are conducted on two real medical datasets, covering different categories of baseline methods with rich evaluation indexes.

Weaknesses
1.	Limited Technical Novelty: The proposed CIB module aims to extract condition-specific drug substructures, but substructure-based modeling has already been explored in prior work. For instance, SafeDrug [1] and MoleRec [2] both model drugs as combinations of molecular substructures with considerations of drug-drug interaction (DDI) safety. Moreover, the use of vector quantization in representation compression is not new — techniques like VQ-VAE [3] are well-established. The overall design appears to rely heavily on empirical tuning rather than principled innovation.
2.	Insufficient Validation of Model Interpretability: While Figure 6 presents a case study on substructure clustering, it lacks expert annotations or supporting biomedical literature, making it difficult to assess whether the extracted substructures are truly pharmacologically meaningful or disease-relevant.
3.	Lack of Pharmacological Alignment in Substructure Extraction: Although Section 3.2 introduces a substructure extraction strategy based on Graph Information Bottleneck theory, the paper does not provide concrete examples of the extracted substructures nor any explanation of their alignment with known pharmacological mechanisms. This weakens the model’s claim of clinical interpretability.
4.	Charts need to be optimized: for example, the background color of “DrugRepresentation and Substructure Capture Modul” in Figure 2 is too light for the reader's intuitive understanding.

[1]	Yang C, Xiao C, Ma F, et al. Safedrug: Dual molecular graph encoders for recommending effective and safe drug combinations[J]. arXiv preprint arXiv:2105.02711, 2021.
[2]	Yang N, Zeng K, Wu Q, et al. Molerec: Combinatorial drug recommendation with substructure-aware molecular representation learning[C]//Proceedings of the ACM web conference 2023. 2023: 4075-4085.
[3]	Van Den Oord A, Vinyals O. Neural discrete representation learning[J]. Advances in neural information processing systems, 2017, 30.

---

> ### Author Rebuttal · Authors · 2025-07-30
>
> # Response to Reviewer Y23r:
>
> Thank you very much for your valuable comments. We are sincerely encouraged by your recognition of our motivation, technical design, and experimental results. Below, we provide point-by-point responses to your suggestions and concerns.
>
> > **W1&Q1:** Limited Technical Novelty?
>
> Thank you for the thoughtful comment. Each module in our framework is carefully designed with domain-specific considerations for both molecular drug structures and clinical patient data.  We acknowledge that prior works, such as SafeDrug [1] and MoleRec [2], also adopt substructure-based modeling and consider DDI safety. However, SafeDrug and MoleRec rely on rule-based fragmentation (e.g., BRICS), which disrupts the natural connectivity of functional groups and fails to capture the synergistic effects of multiple substructures. In contrast, our CIB module learns condition-specific core substructures in a data-driven manner, conditioned on patient health context, ensuring that the extracted motifs are clinically relevant and task-dependent.  Furthermore, motivated by the observation that similarity in current patient visits is often more informative than full-history similarity, we introduce a VQ module to store representative patient–drug interaction prototypes. This module acts as a compact, reusable memory that enhances recommendation in data-limited settings. The VQ also avoids the complexity and instability of high-dimensional variational modeling, making the framework lighter and more stable during training [1,2]. In addition, the discrete latent structure naturally aligns with the CIB principle by encouraging representations that retain only task-relevant information [3].
>
> Overall, SubRec is a framework that unifies condition-aware substructure extraction and discrete patient modeling under a principled architecture. This synergy allows SubRec to achieve SOTA performance while providing chemically and clinically meaningful interpretability, which has not been addressed by prior works.
>
> [1] Mole-BERT: Rethinking Pre-training Graph Neural Networks for Molecules. ICLR, 2023.
>
> [2] Learning Invariant Molecular Representation in Latent Discrete Space. NIPS, 2024.
>
> [3] Conditional Graph Information Bottleneck for Molecular Relational Learning. ICML, 2024.
>
> > **W2:** Insufficient Validation of Model Interpretability.
>
> Indeed, existing works in this field, especially **SafeDrug[1] and MoleRec [2], do not provide explicit demonstrations of the importance of drug substructures. Therefore, our original manuscript did not include such visualizations. In the revised version, following your advice, we have added drug substructure visualization results**, which clearly show that our method captures chemically coherent and clinically relevant motifs. Due to policy restrictions, we are unable to include these figures in the rebuttal, and we kindly ask for your understanding.
>
>
> > **W3:** Lack of Pharmacological Alignment, provide concrete examples of the extracted substructures.
>
> Following your suggestions，indeed, existing works in this field, especially **SafeDrug[1] and MoleRec [2], do not provide explicit demonstrations of the importance of drug substructures. Therefore, our original manuscript did not include such visualizations. In the revised version, following your advice, we have added drug substructure visualization results**, which clearly show that our method captures chemically coherent and clinically relevant motifs. Due to policy restrictions, we are unable to include these figures in the rebuttal, and we kindly ask for your understanding.
>
> Regarding **pharmacological alignment**, we have addressed your concern from two perspectives. First, we present results showing that our model recommends similar drugs for patients with similar diagnostic codes, reflecting alignment with clinical prescribing practices (Table 6). Second, we provide real case studies of the recommended drugs, demonstrating that the extracted substructures correspond to known pharmacological mechanisms. These additions reinforce our claim that the proposed method offers clinically interpretable and pharmacologically meaningful predictions.
>
> | **Category**   | **Patient X**                                                                                                                                               |
> | -------------- | ----------------------------------------------------------------------------------------------------------------------------------------------------------- |
> | **Diagnosis**  | Postconcussion syndrome, Cerebral artery occlusion, Subarachnoid hemorrhage, Acute kidney failure, Retention of urine, Hypertensive chronic kidney disease. |
> | **Procedure**  | Insertion of indwelling urinary catheter, Venous catheterization.                                                                                           |
> | **Medication** | Neomycin, Cefotaxime, Chlorhexidine, Nimodipine, Heparin, Glyceryl trinitrate, Sultiame, Amiodarone, Potassium chloride, Furosemide...                      |
> | **SubRec**     | Neomycin, Cefotaxime, Nimodipine, Heparin, Glyceryl trinitrate, Sultiame, Potassium chloride, Furosemide, Carbamazepine, Mannitol...                        |
> | **Category**   | **Patient Y**                                                                                                                                               |
> | **Diagnosis**  | Subendocardial infarction, Coronary atherosclerosis, Hypertension, Asthma, Hypercholesterolemia.                                                            |
> | **Procedure**  | Insertion of coronary artery stent, heart cardiac catheterization, Coronary arteriography. Insertion of transvenous pacemaker system.                       |
> | **Medication** | Ditazole, Simvastatin, Paracetamol, Practolol, Potassium, Omeprazole, Chloride, Thonzylamine, Tilidine, Sultiame, Oxitriptan, Zafirlukast, Captopril...     |
> | **SubRec**     | Ditazole, Simvastatin, Practolol, Potassium, Chloride, Thonzylamine, Sultiame, Oxitriptan, Zafirlukast, Captopril, Oxyphenisatine...                        |
>
> With regard to the pharmacological alignment analysis of molecular substructures, current research has rarely considered this issue, and thus there are not enough real-world cases for analysis. However, to further address your concern, we have **re-adjusted our model** to specifically target the prediction of **metabolism-mediated diseases**, while focusing on the **active regions** of drugs. Since this is an approximate analysis, we conducted testing in the form of **drug pair inputs**. This setting allows us to better examine whether the extracted substructures correspond to the regions where the drugs exert their effects. Specifically, we utilized a curated dataset of 73 chemicals (perpetrators) known to inhibit metabolic enzymes via specific functional groups, leading to metabolism-mediated DDIs. These chemicals formed 13,786 pairs with other drugs. Using SubRec, we conducted interpretability analysis by comparing the substructures identified by our model with the enzyme-inhibition functional groups reported in the literature.
>
> | **Evaluation Aspect**                   | **Hit-Rate (%)** |
> | --------------------------------------- | ---------------- |
> | **DDI-level Matching**                  | **60.13**        |
> | **Perpetrator-level Matching**          | **87.53**        |
> | **Frequent Functional Groups Matching** | **78.32**        |
>
> [1] Learning motif-based graphs for drug–drug interaction prediction via local–global self attention. Nature machine intelligence. 2024.
>
>
> > **W4:** Charts need to be optimized.
>
> Thank you for this helpful suggestion. In the revised version, we have optimized the figure 2 design by enhancing the color contrast to improve visual clarity and ensure a more intuitive understanding for readers.
>
>
>
> > **Q2:** Lack of citations to relevant literature.
>
> In response to your comments, we have added relevant references.
>
>
> > **Q3:** Does the specific strategy for constructing the codebook in the vector quantization module shift significantly due to changes in the training set?
>
> Thank you for this insightful comment. In our design, the codebook in the vector quantization module is constructed to capture shared latent patterns across the training set, clustering heterogeneous patient–drug interactions into a compact set of prototypes. This design inherently leverages the semantic similarities present across patients and drug interactions, rather than overfitting to individual samples.
>
> Whether the codebook shifts significantly depends primarily on the distribution of the training data. When new training sets share similar distributions with the original data, the learned codebook remains largely stable, as the latent prototypes continue to capture the dominant interaction patterns. Even when novel or previously unseen conditions appear, the codebook still provides meaningful representations because it encodes generalizable structures (Eqs.16-18). Furthermore, its impact is jointly optimized with the overall learning objectives, rather than acting as an isolated component, which further mitigates sensitivity to data shifts.
>
> Therefore, while extreme distributional changes could lead to adjustments in the learned prototypes, under typical conditions the codebook remains stable and continues to support robust and efficient modeling without requiring frequent reconstruction.
>
> ---------------
> We greatly appreciate your insightful and helpful comments, as they will undoubtedly help us improve the quality of our article. If our response has successfully addressed your concerns and clarified any ambiguities, we respectfully hope that you consider raising the score. Should you have any further questions or require additional clarification, we would be delighted to engage in further discussion.

---

> > ### Comment · Reviewer_Y23r · 2025-08-02
> >
> > I would like to thank the authors for their hard work in addressing my concerns. The revisions have clarified the connection between the article's motivation and methodology, and I now have a better understanding of the core contributions. I am particularly impressed with the authors' efforts in *Pharmacological Alignment*, which has led me to reconsider and raise my score.

---

> > > ### Author Response · Authors · 2025-08-03
> > > **Reply to Reviewer Y23r**
> > >
> > > Thank you very much for your kind words and for reconsidering your evaluation based on the revisions we provided. We greatly appreciate your constructive feedback, which helped us improve the clarity of our work.  Thank you again for your valuable input.

---

### Official Review · Reviewer_Ch3Q · 2025-07-05

**Clarity:** 1
**Significance:** 3
**Originality:** 3
**Rating:** 4
**Confidence:** 4

**Summary:**

This paper introduces SubRec, a unified framework that integrates representation learning across both patient and drug spaces. The framework employs a conditional information bottleneck to extract core drug substructures most relevant to patient conditions, thereby improving interpretability and alignment. Additionally, it incorporates adaptive vector quantization to model patient-drug interactions effectively. The proposed approach is validated on the MIMIC-III and MIMIC-IV datasets, demonstrating its efficacy.

**Questions:**

I will consider increasing my score if the authors fully address the concerns outlined in the weaknesses section.

**Ethical Concerns:**

["NO or VERY MINOR ethics concerns only"]

**Final Justification:**

Thank you to the authors for the detailed response. After reviewing the authors' rebuttal and the comments from other reviewers, my concerns have been addressed. Therefore, I increase my score to 4.

**Limitations:**

yes

**Quality:**

2

**Strengths And Weaknesses:**

Strengths:
1. The paper is complete, and the results and figures are well-visualized.
2. The paper is easy to read and follow.

Weaknesses:
1. Unclear Motivation and Novelty: The motivation behind the proposed SubRec is not sufficiently clear, and its novelty remains ambiguous. The authors have not clearly articulated the differences between the proposed method and prior works such as SafeDrug [39] and MoleRec [40]. Specifically: (1) What are the key limitations of SafeDrug and MoleRec that SubRec aims to address? (2) Why is SubRec expected to outperform these prior methods?

2. Challenges in Personalized Medicine: The authors claim that “SubRec is designed to address two key challenges in personalized medicine: (1) the complexity and sparsity of patient historical records, and (2) extracting patient-specific core substructures.” However:
The complexity and sparsity of patient historical records are common challenges in this domain. The authors have not explained why existing methods cannot adequately address these challenges, or how SubRec overcomes them more effectively. This makes the rationale for the proposed method unclear.

3. Graph Information Bottleneck (GIB): The GIB theory referenced in Figure 2 is not explicitly explained. What are the key differences between the proposed method and the following work?

[1] Tailin Wu, Hongyu Ren, Pan Li, and Jure Leskovec. "Graph Information Bottleneck." Advances in Neural Information Processing Systems, 33:20437–20448, 2020.

4. It is unclear how the CIB is integrated into the proposed framework (as shown in Figure 2). A more detailed explanation is required.

5. Lack of Comparisons with Recent SOTA Methods: The paper does not include comparisons with more recent state-of-the-art methods in this field. The following works should be considered for comparison to provide a more robust evaluation:

[1] SeqCare: Sequential training with external medical knowledge graph for diagnosis prediction in healthcare data. Proceedings of the ACM Web Conference 2023, 2819–2830.

[2] Stage-Aware Hierarchical Attentive Relational Network for diagnosis prediction. IEEE Transactions on Knowledge and Data Engineering, 2023.

[3] Beyond Sequential Patterns: Rethinking Healthcare Predictions with Contextual Insights." ACM Transactions on Information Systems, 2025.

---

> ### Author Rebuttal · Authors · 2025-07-30
>
> # Response to Reviewer Ch3Q:
>
> >**W1&W2:** Substructure-based modeling has already been explored in prior work.
>
> Thank you for this comment. We acknowledge that **SafeDrug** [1] and **MoleRec** [2] are important prior works in substructure-based DDI modeling. However, these approaches rely on **rule-based decomposition (like BRICS)** of drugs, which may disrupt the intrinsic connectivity of functional groups, as illustrated in Figure 1. Drug activity usually arises from the **synergistic effect of multiple substructures** rather than isolated fragments. Moreover, meaningful substructure extraction should also account for patient-specific factors, such as comorbidities, which influence prescription decisions (Line 64–72).
>
> To better clarify our contributions, we have added **drug substructure visualization results** in revised version demonstrating that our method captures chemically coherent and clinically relevant motifs. Due to policy restrictions, we are unable to include these figures in the rebuttal, and we appreciate your understanding.
>
> >**W1:** How SubRec overcomes the challenges.
>
> Thank you for your valuable comment. Current research on drug recommendation can generally be divided into two categories: (i) methods that rely on historical patient information, such as COGNet, and (ii) methods that leverage drug substructures, such as SafeDrug and MoleRec. The former often suffers from the complexity and sparsity of longitudinal EHRs, resulting in high computational costs and difficulty in capturing transferable patterns (as shown in the following table). In particular, COGNet, as a typical historical information–based model, exhibits the highest memory consumption. GameNet is another early approach relying on historical data, but it primarily performs lookup-based matching with dynamic memory in key–value form, which falls outside the scope of deep learning models. To address these challenges,  SubRec employs a vector quantization (VQ) module to cluster heterogeneous patient–drug interactions into a compact codebook, mitigating the instability of variational modeling and enabling efficient similarity matching.
>
> |      | Memory (MB) | Para. (M) |
> | ---------- | -------- | ----- |
> | GameNet  | 496 | 0.44  |
> | SafeDrug   | 1716| 0.37 |
> | MoleRec    | 1422 | 0.51  |
> | COGNet     | 3266  | 1.36 |
> | SubRec (codebook size = 32) | 1762| 2.97 |
>
> The latter, represented by approaches such as SafeDrug and MoleRec, typically uses rule-based method (e.g., BRICS) to split molecules into fragments, which disrupts the natural connectivity of functional groups and fails to reflect the synergistic effect of multiple substructures.  Here, we would like to emphasize that while it is true that clinicians typically rely on medical knowledge and pharmacological guidelines, **the molecular structure remains a fundamental determinant** of a drug’s pharmacodynamics and mechanism of action[1,2]. 1) Many drugs exert their therapeutic effects through **functional substructures**, or pharmacophores. For example, _simvastatin_ and _atorvastatin_ share a common pharmacophore that inhibits HMG-CoA reductase, making both effective for lowering cholesterol[3]. 2) Moreover, the latest research has present that  **substructures also contribute significantly to potential adverse drug interactions**. Shared or reactive substructures can lead to overlapping metabolic pathways or toxicity profiles, which are crucial in assessing the safety of drug combinations[4,5].  Therefore,  we incorporates an improved CIB to extract core drug substructures conditioned on patient health context, ensuring that the learned representations remain both concise and clinically relevant. The VQ mechanism further supports CIB by enforcing a discrete latent structure that preserves only the most task-relevant information. Through this design, SubRec overcomes the limitations of prior methods, achieving a better balance between predictive accuracy, robustness, and interpretability.
>
> Indeed, existing works in this field, especially **SafeDrug[1] and MoleRec [2], do not provide explicit demonstrations of the importance of drug substructures. Therefore, our original manuscript did not include such visualizations. In the revised version, following your advice, we have added drug substructure visualization results**, which clearly show that our method captures chemically coherent and clinically relevant motifs. Due to policy restrictions, we are unable to include these figures in the rebuttal, and we kindly ask for your understanding.
>
> [1] Why drugs fail—a study on side effects in new chemical entities. Nat. Rev. Drug Disc.  2005
> [2] Pharmacophore modeling in drug discovery. Nat. Rev. Drug Disc. 2016
> [3] Large-scale prediction and testing of drug activity on side-effect targets. Cell. 2012
> [4] Learning motif-based graphs for drug–drug interaction prediction via local–global self attention. Nat mach Intell. 2024
> [5] Emerging drug interaction prediction enabled by a flow-based graph neural network with biomedical network. Nat Comp Sci. 2023
>
> >**W2:** The complexity and sparsity of patient historical records.
>
> Thank you for your insightful comment. We would like to further elaborate on how our framework addresses the complexity and sparsity of patient historical records.
>
> First, regarding complexity, as noted in our response to the previous question, modeling the full visit history of patients can lead to significant memory consumption, especially in deep learning frameworks. To mitigate this, we designed a VQ module to encode heterogeneous patient–drug interactions into a compact codebook. This representation reduces computational overhead while retaining valuable information from historical visits.
>
> Second, concerning sparsity, our dataset is split based on patients, but the number and frequency of visits vary significantly. Some patients have rich historical records, while others have only a few visit. This inconsistency introduces sparsity into the training process. In the extreme case of one-shot scenarios (i.e., patients with only one visit), we evaluate model performance and report the results in Figure 4, with detailed descriptions in Appendix E.3.
>
> In addition, we considered a single-drug recommendation scenario, where each visit results in only one prescribed drug. This setting provides limited contextual information from longitudinal history. Our evaluation on the MIMIC-III dataset shows that models heavily dependent on patient history (e.g., COGNet) perform worse under this constraint. In contrast, structure-based models, MoleRec and SafeDrug, maintain superior performance, as they can capture intrinsic associations between molecular substructures and clinical conditions. The GIB module further strengthens this by extracting condition-specific drug substructures, allowing SubRec to remain effective even in sparse or low-information settings, and outperforming other baselines.
>
> | SubRec   | 0.1620 ± 0.1641  |
> | ------- | --------- |
> | MoleRec  | 0.1269 ± 0.1108  |
> | COGNet   | 0.1037 ± 0.0086  |
> | SafeDrug | 0.0755 ±  0.0043 |
> | GameNet  | 0.0521 ± 0.0032  |
>
> >**W3:** What are the key differences between the SubRec and GIB (NIPS 2020) ?
>
> Recently, information bottleneck (IB) theory (Arxiv., 2000) has been applied to learning significant subgraphs of the input graph for explainable GNNs (NIPS 2020, CVPR 2022; 2020; ICML 2022), which provides a principled approach to determine which aspects of data should be preserved and which should be discarded (AAAI 2021). However, directly applying GIB in the context of drug recommendation is non-trivial. Traditional GIB-based methods learn the core subgraph solely from the input graph itself, while in drug recommendation, the task requires **considering patient-specific conditions** to recommend optimal drug combinations. Therefore, our framework learns a **condition-specific core subgraph** $\mathcal{\widetilde{G}}_{\mathrm{IB}}$ that maximizes the mutual information between drug substructures and the prediction target $Y$ (Eqs. 6–7).
>
> Moreover, our approach aims to go beyond existing GIB applications by integrating precise substructure-level reasoning with longitudinal patient records in an organic manner. This joint modeling allows SubRec to leverage both molecular structural knowledge and patient-specific clinical context, carefully balancing their contributions to achieve high predictive performance while simultaneously maintaining a low DDI rate.
>
> >**W4:** How the CIB is integrated into the proposed framework.
>
> The framework models patient representations for prediction from three complementary aspects.
> First, by leveraging drug substructure information and the conditional drug CIB mechanism, the model focuses on extracting condition-specific core features from drug substructures while suppressing irrelevant information. This enables the generation of targeted molecular representations for all candidate drugs, thereby improving prescription accuracy. Second, by incorporating historical information, the framework employs a VQ module to construct a compact codebook that reduces complexity and efficiently encodes heterogeneous patient–drug interactions. Finally, through the interaction between condition-aware substructures and quantized patient states, the model produces reliable recommendation outcomes while maintaining a low DDI rate.
>
>
> >**W5:** Lack of Comparisons with Recent SOTA Methods.
>
> Thank you for your suggestion. We have added the mentioned baseline model (Jaccard Index) in the revised version. The results show that our method maintains a clear advantage on the **MIMIC-III** dataset, while on **MIMIC-IV** it performs slightly lower than **BSP**, **SeqCare**, and **HAR**.
>
> |      | MIMI-III | MIMI-IV |
> | ----- | ------ | ----- |
> | HypEHR  | 0.5139   | 0.4837|
> | MedPath | 0.4997  | 0.4547|
> | HAR     | 0.5049 | 0.4789 |
> | SeqCare | 0.5097| 0.4827|
> | BSP     | 0.5375 | 0.4986 |
> | SubRec  | 0.5585  | 0.4635 |

---

> > ### Author Response · Authors · 2025-08-04
> > **Looking Forward to Your Feedback and Support**
> >
> > Dear Reviewer,
> >
> > We sincerely appreciate the time and effort you have dedicated to reviewing our manuscript. We recently submitted a detailed response addressing all the concerns you raised, including additional experiments, clarifications, and revisions throughout the paper. **We would like to kindly ask whether our responses have adequately resolved your concerns. If there are any further issues or areas that require additional clarification or improvement, please do not hesitate to let us know. We are more than willing to make further adjustments to ensure the work meets your expectations.**
> >
> > If our replies have addressed your concerns to your satisfaction, we would be grateful if you could consider reflecting this in your evaluation by providing a positive score. Thank you once again for your constructive feedback and for helping us improve the quality of our work.
> >
> > Warm regards,
> >
> > The authors

---

> > > ### Author Response · Authors · 2025-08-05
> > >
> > > Dear Reviewer Ch3Q ,
> > >
> > > We sincerely appreciate the time and effort you have devoted to reviewing our manuscript. Previously, we submitted a detailed response addressing all the concerns you raised, including additional experiments, clarifications, and revisions throughout the paper.
> > >
> > > As of now, **we have not yet received your feedback regarding whether our responses have adequately resolved your concerns.** We would like to kindly follow up to confirm if our replies have sufficiently addressed your points. If there are still any unresolved issues or areas requiring further clarification, please do not hesitate to let us know—we remain fully committed to making additional improvements to ensure the work meets your expectations.
> > >
> > > If our replies have satisfactorily resolved your concerns, we would be truly grateful if you could consider reflecting this in your evaluation by providing a positive score. Your feedback is of great significance to us, and your support would mean a great deal.
> > >
> > > Thank you once again for your constructive comments and for contributing to the improvement of our work. We sincerely look forward to your response.
> > >
> > > Warm regards,
> > >
> > > The Authors

---

### Note · Authors · 2025-08-12

Dear ACs and Reviewers,

We extend our sincere gratitude for the time and effort you have dedicated to review our manuscript. We greatly enjoy the in-depth discussions with the reviewers and appreciate their valuable feedback.

| **Reviewer** | **Main Concerns** | **Response**  | **Outcome** |
| --- | ---- | --- | --- |
|`Ch3Q` | Novelty of Motivation module; unclear rationale; differences from GIB (NIPS 2020); missing baselines.                                                                                               | Clarified differences from prior work; explained why SubRec effectively addresses potential issues; added experiments on complexity/sparsity settings; added baselines| No feedback. |
|`Y23r`| Concerns on technical novelty; differences from previous works; model interpretability; lack of pharmacological alignment analysis; chart optimization.    | Detailed technical contributions and differences from SafeDrug and MoleRec; added interpretability analyses; conducted case studies on metabolism-mediated diseases for pharmacological alignment; optimized charts; added references and VQ module explanation. | Score raised.|
|`3FcV` | Model complexity; ablation studies; internal connections between modules; noise injection mechanism.                                                                                                   | Compared runtime with baselines; reduced model runtime via various strategies; clarified ablation study results; explained module interconnections; provided theoretical proof for noise injection.| Positive feedback.|
|`2y9i`| Interest in work but requested additional explainability analysis; Analysis in real-world settings; confusion on Equ. 8 and 9; unclear motivation for discrete embeddings. | Explained design motivations of each module; justified discrete embeddings for memory efficiency and leveraging historical info; clarified technical contributions and module interconnections; added theoretical derivations for Equ. 8 and 9; provided supporting results and analyses.| Concerns resolved.|

---

Thank you very much for your patience and the considerable effort you have devoted to our work. **We truly value this opportunity to present our work at NeurIPS and sincerely** hope to earn the AC’s understanding and support.

Best regards,

Authors

---

### Decision · Program_Chairs · 2025-09-17

**Decision:**

Accept (poster)

**Comment:**

This paper presents and evaluates a deep learning approach to provide patient treatment recommendations based on all of patient EHR data (history as well as properties of current visit symptoms, diagnoses, prescriptions, labs, vitals, etc.), molecular structures of drug treatments being considered, and potential drug combinations and drug-drug interactions.  The reviewers and authors had a robust discussion, the clarifications and recommendations of which should be incorporated into the paper to make it stronger, including new results that the authors mentioned that were obtained during the discussion phase.  Reviewers became more positive during the discussion, but frankly the paper is still very much on the borderline.  On the negative side, we are still a long way from clinicians using AI recommendations to decide on treatment combinations for patients (especially from deep neural networks given the lack of transparency), it is unclear how successful use of detailed molecular structures will be at this level (in contrast to the drug design or individual adverse event discovery phases), I'm not sure I agree that drug activity is the result of many different substructures (I think of "substructure" as including a 3D arrangement of components such as hydrophobic groups, hydrogen acceptors/donors, etc.; maybe if we include multiple "activities" including ADEs, side effects, unintended positive effects, etc then the claim is true), and the methods here are not especially innovative.  Nevertheless, on the positive side, we should indeed be working toward a foundation model for all of health, so we do indeed need methods now incorporating EHRs, molecular structures (and even protein structures, microbiome, etc.), and this paper is a step in that direction.  Also on the positive side, we do indeed desperately need to consider effects of combinations of multiple drugs rather than single drug effects.  And the "reverse" approach of predicting structure from the EHR data, rather than vice-versa, is a nice innovative approach motivated by the information bottleneck argument.  The deep learning architecture is sound, as is the evaluation methodology.